# Comprehensive Therapeutic Approaches to Tuberculous Meningitis: Pharmacokinetics, Combined Dosing, and Advanced Intrathecal Therapies

**DOI:** 10.3390/pharmaceutics16040540

**Published:** 2024-04-14

**Authors:** Ahmad Khalid Madadi, Moon-Jun Sohn

**Affiliations:** 1Department of Biomedical Science, Graduate School of Medicine, Inje University, 75, Bokji-ro, Busanjin-gu, Busan 47392, Republic of Korea; khalidmadadi@yahoo.com; 2Department of Neurosurgery, Neuroscience & Radiosurgery Hybrid Research Center, College of Medicine, Inje University Ilsan Paik Hospital, 170, Juhwa-ro, Ilsanseo-gu, Goyang City 10380, Republic of Korea

**Keywords:** intrathecal, intraventricular, anti-TB drugs, tuberculous meningitis, prolonged drug delivery

## Abstract

Tuberculous meningitis (TBM) presents a critical neurologic emergency characterized by high mortality and morbidity rates, necessitating immediate therapeutic intervention, often ahead of definitive microbiological and molecular diagnoses. The primary hurdle in effective TBM treatment is the blood–brain barrier (BBB), which significantly restricts the delivery of anti-tuberculous medications to the central nervous system (CNS), leading to subtherapeutic drug levels and poor treatment outcomes. The standard regimen for initial TBM treatment frequently falls short, followed by adverse side effects, vasculitis, and hydrocephalus, driving the condition toward a refractory state. To overcome this obstacle, intrathecal (IT) sustained release of anti-TB medication emerges as a promising approach. This method enables a steady, uninterrupted, and prolonged release of medication directly into the cerebrospinal fluid (CSF), thus preventing systemic side effects by limiting drug exposure to the rest of the body. Our review diligently investigates the existing literature and treatment methodologies, aiming to highlight their shortcomings. As part of our enhanced strategy for sustained IT anti-TB delivery, we particularly seek to explore the utilization of nanoparticle-infused hydrogels containing isoniazid (INH) and rifampicin (RIF), alongside osmotic pump usage, as innovative treatments for TBM. This comprehensive review delineates an optimized framework for the management of TBM, including an integrated approach that combines pharmacokinetic insights, concomitant drug administration strategies, and the latest advancements in IT and intraventricular (IVT) therapy for CNS infections. By proposing a multifaceted treatment strategy, this analysis aims to enhance the clinical outcomes for TBM patients, highlighting the critical role of targeted drug delivery in overcoming the formidable challenges presented by the blood–brain barrier and the complex pathophysiology of TBM.

## 1. Introduction

Tuberculous meningitis (TBM), the most severe manifestation of extrapulmonary tuberculosis (TB), is rare but carries a significant global impact [1]. It affects 1–5% of TB cases worldwide, constituting approximately 13.91% of meningitis cases (WHO Global Tuberculosis Report 2020) [2,3,4]. In 2019 alone, TBM resulted in an estimated 78,200 (95% UI; 52,300–104,000) adult deaths, accounting for 48–50% of incident cases globally, representing a significant proportion of incident TBM cases [5,6]. Vulnerable populations, including children under 5 and immunocompromised individuals (HIV-1 co-infected patients), face increased risks [7,8,9]. Delayed diagnosis and treatment can lead to fatality rates exceeding 50% and severe neurological complications [10]. Despite its lower prevalence, TBM carries significant mortality and morbidity rates, particularly among HIV-1 co-infected individuals (50%) and children (19.3%) [11,12,13,14]. TBM, accounting for about 20% of childhood TB mortality, often results in neurological complications in over half of survivors [11,15].

TBM poses significant challenges in treatment due to the limited penetration of drugs into the central nervous system (CNS) and the emergence of drug-resistant strains. Our emphasis lies on prolonged intrathecal (IT) drug administration to bypass the barriers and mitigate systemic side effects and risks linked to repetitive IT injections while improving therapeutic outcomes by ensuring sufficient anti-TB drug concentrations at the site of infection. Current TBM treatment strategies involve long-term and complex antimicrobial therapy, often resulting in adverse effects and patient non-compliance [4,12,16]. Determining the optimal combinations, durations, doses, and frequencies of drug regimens for TBM remains uncertain, with insufficient evidence to guide empiric treatment regimens for TBM [11]. The blood–brain barrier (BBB) complicates matters by impeding drug delivery to the CNS, necessitating alternative administration methods to ensure optimal drug concentration in the cerebrospinal fluid (CSF) and brain [17].

The IT drug delivery, involving the direct injection of substances into the thecal sac housing CSF, represents a promising approach for drug administration. This strategy enables the attainment of elevated concentrations of therapeutic agents within the CNS, concurrently mitigating off-target exposure and the associated toxicity [18,19]. Despite its potential, IT therapy presents risks and limitations, encompassing chemical ventriculitis or meningitis, seizures, local adverse events, and infections [20,21]. Historically, IT/intraventricular (IVT) administration of isoniazid (INH) and rifampicin (RIF) for TBM, documented since 1955 and 1976, respectively, shows promise in refractory cases resistant to oral treatments, allowing for direct CNS targeting, enabling reduced dosages, minimizing side effects, and improving therapeutic outcomes without reported serious side effects. The initial documented use of IVT-RIF involved a TBM patient experiencing relapse with tuberculoma complications [22]. Cases involving IVT-RIF and systemic anti-TB drugs demonstrate high efficacy and safety without documented toxicity [23,24]. The treatment with IT amikacin (AMK) and levofloxacin (LVX) resulted in successful clinical and microbiological outcomes even in the case of multidrug-resistant TBM (MDR-TBM) [25]. The current WHO guidelines for treating TBM are derived from those established for pulmonary tuberculosis. These guidelines recommend a treatment regimen comprising two months of RIF, INH, pyrazinamide (PZA), and ethambutol (ETB), followed by a continuation phase of up to ten months consisting of RIF and INH for all TBM patients [26]. However, determining the optimal duration of treatment lacks a consensus based on evidence, resulting in variations in treatment duration observed between different countries [27,28]. The global prevalence of rifampicin/multidrug-resistant TBM (RR/MDR-TBM) is exhibiting an upward trend, leading to catastrophic prognostic implications, including a mortality rate exceeding 80% [29,30,31]. Therefore, early diagnosis of TBM and evaluation of its drug resistance and customization of treatment regimens to achieve a high concentration of anti-tuberculosis (RIF or INH) in CSF are essential for effectively managing RR/MDR-TBM [32].

This review meticulously synthesizes clinical experiences, case reports, and the pharmacokinetics–pharmacodynamics (PK-PD) of RIF and INH within the CSF, spotlighting the critical gaps in achieving sustained drug concentrations, exploring alternative drug formulations, and refining treatment regimens. The review emphasizes the potential of the potential of innovative IT drug delivery methods, such as osmotic pumps and nanoparticles (NP)-laden hydrogels, which promise enhanced and prolonged drug release directly into the CSF. These advancements aim to circumvent the limitations posed by traditional administration routes, offering further research to optimize dosage regimens and improve TBM treatment strategies. TBM persists as a formidable challenge within the spectrum of infectious diseases, primarily due to its severe impact on the central CNS and the intricate barriers to effective treatment. Among the myriad of hurdles, the PK dynamics of anti-TB drugs within the CSF stand out as a critical determinant of therapeutic success. This comprehensive review investigates the physiological and pharmacological factors influencing PK of anti-TB drugs in the CSF, emphasizing the theoretical approach and strategy required to optimize treatment for this devastating condition.

### 1.1. Physiology of BBB and CSF

A thorough grasp of the CSF system is vital for understanding how substances within it behave and the clinical implications for drug delivery. CSF serves essential roles in brain protection, maintaining balance, and regulating neuronal function. Numerous studies have demonstrated the critical importance of normal CSF production, circulation, and absorption for typical brain development and function.

In adults, CSF volume is approximately 150 mL, with about 25 mL in brain ventricles, 50 mL in cerebral subarachnoid space, and 75 mL in spinal subarachnoid space. Daily CSF production ranges from 400–600 mL, predominantly by the choroid plexuses of lateral ventricles. CSF composition includes water (99%), proteins at low concentrations, ions, neurotransmitters, and glucose. Its production is finely controlled by the autonomic nervous system and neuropeptides like dopamine and atrial natriuretic peptides. Sympathetic nervous system activity reduces CSF secretion, while cholinergic system activation enhances it. Following production in lateral ventricles, CSF flows through the interventricular foramen of Monro and aqueduct into the third and fourth ventricles, respectively. Subsequently, it passes through the median (foramen of Magendie) and lateral (foramina of Luschka) apertures in the fourth ventricle into the subarachnoid space at the base of the brain, then over the brain’s convexity and along the spinal cord [33].

Recent studies have unveiled additional pathways involved in CSF and solute movement throughout the CNS. In an experimental study by Iliff et al., conducted in mice, it was observed that CSF traverses intracortical periarterial spaces akin to blood flow. Subsequently, CSF moves through perivascular astrocytic endfeet mediated by aquaporin-4 (AQP4) water channels, entering the parenchyma where it merges with interstitial fluid. CSF then exits the parenchyma through AQP4 channels, returning to the subarachnoid space via perivenous spaces. From there, it can follow two routes: drainage through arachnoid granulations into venous sinuses or exit via the lymphatic system. This insight was gleaned from injecting fluorescently labeled tracers through the cisterna magna of mice and tracking flow in real-time with in vivo two-photon microscopy. Both small (3 kilodaltons, kDa) and large (2000 kDa) tracers could traverse the perivascular space of penetrating arteries and arterioles. However, solute entry into the interstitial space was size-dependent, with small molecules dispersing quickly throughout the brain interstitium and larger molecules concentrating in the perivascular space. This discrepancy is attributed to the “sieving effect” of perivascular astrocytic endfeet, acting as a physical barrier with gap widths of approximately 20 nanometers. Moreover, clearance of solutes occurs from the interstitial space along perivenous spaces to the subarachnoid space and beyond. Polarized AQP4 water channels on astrocyte endfeet facilitate CSF flow into and through the interstitium while clearing fluid and solutes from the interstitial space [34]. AQP4 constitutes a crucial component of the glymphatic system, characterized by 0.3- to 0.6-nanometer pores selectively permeable to water, widely expressed on capillary-facing astrocyte endfeet along perivascular spaces. However, the precise mechanism by which AQP4 facilitates solute elimination and bidirectional delivery to the parenchyma remains poorly understood [35]. These experiments introduced the concept of the “glymphatic system”, a waste clearance system of the CNS utilizing perivascular channels formed by astroglial endfeet, responsible for removing amyloid β and other waste products from the CNS [34]. Traditionally, CSF absorption was thought to occur primarily in arachnoid villi along superior sagittal and intracranial venous sinuses and around spinal nerve roots. However, recent studies propose additional arachnoid pathways as major sites of CSF absorption, including passage through brain parenchyma, lymphatics near the cribriform plate, or perineural sheaths of cranial nerves [33].

### 1.2. Physiological Factors Influencing IT-PK

The blood–brain and blood–CSF barriers: The main challenge in treating TBM lies in the BBB and the blood–CSF barrier (BCSFB), intricate networks of lipid layers designed to shield the CNS from potential threats. These barriers, while vital for neuroprotection, significantly impede the entry of anti-TB drugs to the sites of infection within the CNS, necessitating higher systemic doses that often come with increased toxicity risks. The delicate balance between protecting the brain and allowing therapeutic agents access is a critical consideration in developing effective TBM treatments.

The efficacy of anti-TB drugs in treating TBM is contingent upon achieving a specific exposure level at the infection site, particularly within the CSF. Determining the appropriate dosing regimen for IT administration necessitates a consideration of various factors. These factors include the BBB and BCSFB, the size of the CSF system, CSF volume, the site of antibiotic administration (IT or IVT), and drainage volume. The BBB and BCSFB encompass lipid layers enveloping the CNS to protect the brain from harmful agents circulating in the bloodstream. These barriers play a crucial role in restricting the entry of antibiotics to the infection site during the treatment of meningitis or ventriculitis.

The role of inflammation in drug permeability: Inflammation, a characteristic response to TBM, disrupts the BBB’s integrity, variably enhancing the CSF’s permeability to antibiotics, especially those with hydrophilic properties [36,37,38]. This disruption, though potentially beneficial in increasing drug penetration, introduces a layer of unpredictability in treatment outcomes. This effect is more pronounced in meningitis than in ventriculitis. Neurosurgical procedures may further disrupt the BBB, potentially increasing systemic antibiotic penetration [39]. However, the extent of penetration during inflammation is unpredictable. The extent of BBB disruption and its impact on drug delivery efficacy underscore the need for adaptable and precise dosing regimens that can account for the dynamic nature of the disease.

The CSF compartment: A challenge of volume and flow. The CSF compartment’s complexity adds another layer of difficulty in achieving optimal drug delivery. Variability in the compartment’s size, influenced by factors such as age and disease, affects the distribution volume of administered drugs. Furthermore, the non-uniform distribution and directional flow of CSF necessitate careful consideration of the site of antibiotic administration. The choice between IT and IVT dosing can significantly impact the distribution and efficacy of anti-TB drugs within the CNS.

The CSF compartment’s size, crucial for determining the distribution volume post-BBB penetration, shows significant interindividual variability, influenced by age, ventricular and subarachnoid space dimensions, and underlying diseases. The dimension of the CSF compartment, a pivotal determinant for the volume of distribution (V_CSF_) in IT dosing following BBB penetration, demonstrates notable variability. The V_CSF_ encompasses the four ventricles, an aqueduct, basal cisterns, and the subarachnoid space. Interindividual variations in V_CSF_ size are influenced by factors such as age, widths of ventricles and subarachnoid space, dimensions of the spinal canal, and underlying diseases, all contributing to the diversity in this critical parameter [40]. Studies have documented varying mean CSF volumes across different conditions, with the presence of clots potentially reducing this volume. Numerous studies have reported mean CSF sizes in healthy adults ranging from 250 to 326 mL [41,42,43], in communicating hydrocephalus patients at 488 mL, and in non-communicating hydrocephalus patients at 593 mL [43]. The presence of clots in ventricles or basal cisterns may result in a reduction in V_CSF_ size. Despite the intricate nature of the CSF compartment, its distribution exhibits non-uniformity [40,44]. Approximately two-thirds of the CSF is generated by the choroid plexus, with the remaining one-third originating from the extracellular space of the brain and spinal cord. CSF equilibration relies on oscillations induced by heartbeat and respiration. The overall flow of CSF occurs from the ventricles to the cisterna magna, cerebral convexities, and the spinal canal. The concentrations of antibiotics obtained through lumbar puncture and extra ventricular drain sampling may vary. Following entry into the CSF, antibiotics can diffuse into the extracellular fluid of the brain and spinal cord due to the absence of a tight barrier, contrasting with the bulk flow of CSF [40].

PK considerations for effective drug delivery: Effective management of TBM requires not only overcoming physiological barriers but also considering the PK landscape of drug administration. The drainage volume of CSF, influenced by factors such as meningitis-induced production changes and external drainage systems, plays a crucial role in the clearance of antibiotics from the CSF. Understanding these dynamics is essential for timing IT doses to maximize therapeutic efficacy while minimizing the risk of drug resistance.

Given the complexity of physiological factors affecting IT-PK, it becomes increasingly clear that a one-size-fits-all approach to TBM treatment is insufficient. The advent of advanced drug delivery mechanisms, such as nanoparticle-infused hydrogels and osmotic pumps, offers promising avenues for achieving sustained drug release within the CNS. These innovations, coupled with a deeper understanding of the physiological and pharmacological factors at play, pave the way for tailored, effective treatment strategies that can navigate the complex terrain of TBM therapy.

Selecting the administration site for IT doses is of paramount importance due to the non-homogeneous characteristics of the CSF compartment and the directional flow of CSF from the ventricles. Discrepancies in drug concentrations are noted across the ventricular, cisternal, and lumbar regions within the CSF compartment [45]. As IT doses are frequently employed for ventriculitis, ensuring optimal antibiotic exposure in the ventricles becomes essential, making IVT administration preferable. In contrast to intravenous (IV) administration, where lumbar CSF concentrations generally surpass ventricular CSF concentrations for most drugs [36], IVT dosing guarantees distribution throughout the CSF compartment unless impeded by factors such as bleeding. The intraluminal or intraspinal administration of antibiotics leads to decreased concentrations in ventricular CSF attributed to the net flow of CSF originating from the ventricles [46]. This phenomenon results in an uneven distribution within the CSF space and the possibility of insufficient drug levels in the ventricles. PK data endorse IVT dosing, especially in the management of ventriculitis, despite the absence of current clinical evidence supporting this preference [38]. The drainage volume, indicating the daily CSF elimination in mL/day, significantly influences the clearance of antibiotics from the CSF. Keeping the external drain at a consistent level facilitates CSF excretion during elevated intracranial pressure surpassing drain pressure, and any changes in drain levels may impact drainage volume. Research highlights the crucial role of drainage volume in determining IT antibiotic doses [47,48,49]. CSF production, exhibiting circadian variation, reaches a minimum of 12 ± 7 mL/h around 6 p.m. and a nightly peak of 42 ± 2 mL/h around 2 a.m. in both healthy volunteers and patients with external ventriculostomies [50,51]. Meningeal inflammation reduces CSF production, affecting drainage volume and antibiotic clearance, emphasizing the importance of timing IT doses [36].

In summary, the strategy for optimizing TBM treatment is fraught with challenges, from considering physiological barriers to understanding the intricacies of ITPK. By embracing a multifaceted approach that combines PK insights with innovative drug delivery systems (DDS), we can move closer to turning the tide against this devastating disease, offering hope for improved outcomes in the fight against TBM.

### 1.3. Advancing IT Therapy for TBM: Beyond Physiological Barriers to PK Optimization

PK factors affecting IT therapy: The journey of anti-TB drugs from the systemic circulation to the CSF is influenced by a myriad of PK variables. The clearance of drugs from CSF to blood is intricately affected by the drugs’ inherent characteristics. This process is governed by bulk flow, retrograde diffusion across the blood–CSF and BBB, and active transport mechanisms. The dynamics of these processes underscore the nuanced approach needed to ensure that therapeutic agents reach their target sites within the CNS effectively.

The IT-PKs of anti-TB drugs are greatly impacted by the drugs’ intrinsic characteristics and their systemic administration. The process of clearing drugs from the CSF to the blood, indicated as CLCSF to Blood, is profoundly influenced by the drugs’ properties. The elimination process is dependent on several drug-specific factors. Key elements that influence CLCSF to Blood include (1) bulk flow, which is equivalent to the rate of CSF production, (2) retrograde diffusion across the blood–CSF and BBB, and (3) active transport mechanisms [52]. Bulk flow primarily facilitates the removal of large hydrophilic molecules, leading to decreased clearance rates in patients with hydrocephalus [40,53]. In contrast, small to moderately lipophilic molecules predominantly are cleared through passive elimination via retrograde diffusion across the barriers [54].

The Role of Drug Properties in CSF Penetration: The PK behavior of anti-TB drugs in the CSF is significantly dictated by their hydrophilic or lipophilic nature, molecular weight (MW), and protein binding affinity. Hydrophilic antibiotics face challenges in penetrating the CSF, highlighting the necessity for strategic dosing to circumvent these limitations. Conversely, lipophilic drugs, with their capacity for easier equilibration and membrane binding, present a different set of considerations for achieving optimal CSF distribution. The impact of drug characteristics on treatment efficacy illuminates the path toward tailored therapeutic strategies that account for these PK characteristics.

Variations in BBB permeability and CSF volume distribution are linked to dissimilarities in the characteristics of hydrophilic and lipophilic antibiotics. Hydrophilic antibiotics demonstrate restricted entry into the CSF, with the CSF volume of distribution determined by the combination of CSF volume distribution and the portion of brain extracellular space equilibrating with CSF [55,56]. Conversely, lipophilic drugs, facilitated by easier equilibration and membrane binding, generally exhibit a greater CSF volume of distribution when compared to hydrophilic drugs. In instances of meningitis, inflammation-induced BBB penetration is more prominent for hydrophilic antibiotics, whereas lipophilic antibiotics exert a minimal impact [38]. The entry into the CSF is markedly affected by both molecular weight and protein binding [57]. In instances where the barrier is intact, only the unbound fraction can traverse the CSF, as binding proteins generally exhibit minimal crossing of the barriers [57]. Compounds characterized by low protein binding display an increased unbound fraction, thereby promoting improved penetration into the CNS, with CSF protein binding generally being lower in comparison to serum/plasma [58]. Furthermore, molecular weight serves as a decisive factor in BBB penetration, with substances featuring larger molecular weights typically experiencing restricted penetration.

Optimizing treatment through combined IT and IV administration: The strategic combination of IT and IV therapies has been identified as a crucial approach for overcoming the challenges posed by the BBB and enhancing drug levels within the CSF. This integrated method not only facilitates higher antibiotic concentrations in the CSF but also mitigates the risks of bacterial resistance and disease relapse. The synergy between systemic and local drug administration, leveraging the strengths of both approaches, represents a forward-thinking strategy in the management of MDR-CNS infections.

The combined use of IT and IV treatments results in slightly higher antibiotic levels in CSF compared to IVT therapy alone, which is crucial for managing MDR-CNS infections [59]. Simultaneous IVT and IV administration of antibiotics prevent low concentrations in brain compartments, reducing the likelihood of resistant bacteria selection and relapse [60]. Concurrent systemic antibiotic therapy, utilizing the same or alternative effective agents, is strongly endorsed and backed by numerous studies showing successful IT antibiotic administration alongside IV therapy [40]. Simultaneous IV and local administrations are deemed optimal, considering CSF circulation and obstruction [48]. However, the effect of simultaneous IV dosing on CSF-PK for all antibiotics remains uncertain [40,52]. While most patients receive a combination of IV and IT doses, evidence supporting a shift to IT-only regimens is lacking. The lack of data solely focusing on IT drug administration is attributed to the necessity of combining IT and IV therapy for optimal therapeutic outcomes [40]. In CNS infections caused by MDR pathogens, the combination of IVT and systemic antimicrobial therapy is acknowledged as potentially life-saving [40].

The imperative for PK research and innovation: Despite the advancements in understanding the PKs of IT therapy, significant gaps remain in our knowledge, particularly regarding the optimal integration of systemic and local drug administration. The lack of comprehensive data focused solely on IT drug delivery underscores the urgent need for research aimed at refining dosage regimens and exploring novel therapeutic interventions. The potential of emerging technologies, such as nanoparticle-infused hydrogels and osmotic pumps, in achieving sustained drug release within the CNS offers a promising avenue for future exploration.

Pioneering a new era in TBM treatment: The effective management of TBM at the intersection of PKs and advanced drug delivery technologies presents a multifaceted challenge. However, it also offers an opportunity to significantly improve patient outcomes. By harnessing insights into drug properties, systemic administration, and innovative delivery methods, we can develop a comprehensive framework for TBM treatment that transcends traditional barriers. This integrated approach not only promises to enhance the precision and efficacy of therapy but also marks a step towards a new era in the battle against TBM, one that is characterized by informed, patient-centric, and outcome-driven strategies.

## 2. Materials and Method

This comprehensive review synthesizes findings from electronic scientific databases, including PubMed, Google Scholar, and Science Direct, focusing on English-language articles related to the use of “Intrathecal or Intraventricular Rifampicin” and “Intrathecal or Intraventricular Isoniazid” in the treatment of TBM. Our initial search yielded 3277 articles, from which 169 were selected through database searches. This pool was further enriched by manually adding references found in recent research articles, systematic reviews, meta-analyses, clinical trials, randomized controlled trials, cohort studies, clinical experiences, and case reports dating from 1958 to the current day. After a thorough screening for title relevance and the removal of duplicate entries, 3108 articles were excluded. The references of the remaining literature were manually checked for completeness, ensuring the inclusion of all pertinent information regarding the IT and IVT treatment modalities for TBM. The inclusion criteria were English-language publications and original research, excluding citations, patents, non-English articles, and abstracts lacking data.

The methodology of this review follows a thematic structure outlined as follows: First, the introduction is presented, followed by an examination of the physiological factors influencing IT-PKs, and subsequently, the advancing IT therapy for TBM. Next, the results obtained from the literature search are detailed. Following this, an analysis of the PKs of systemic anti-TB drugs in CSF is conducted. Additionally, a comparative analysis is undertaken between combined systemic dosing and IT/IVT administration alone. The general aspects of IT/IVT drug administration are then discussed, followed by an exploration of therapeutic drug monitoring. Furthermore, clinical experiences concerning IT/IVT anti-TB treatment are reviewed, along with an evaluation of adverse events and the associated risks of IT/IVT drug administration. Strategies for optimizing therapeutic regimens for IT anti-TB therapy are explored, including discussions on strategies for IT-prolonged drug delivery, such as the clinical application of an osmotic pump and nanoparticle-laden hydrogel.

## 3. Results

### 3.1. PK of Systemic Anti-TB Drug in CSF: General Overview, Importance, and Challenges

Understanding the PKs of first-line anti-TB drugs is essential for the effective management of TB, highlighting the significance of absorption, distribution, metabolism, and excretion processes. Accurate PK knowledge is critical to prevent suboptimal drug levels, thus reducing the risk of treatment failure, mortality [61], and adverse effects, such as drug-induced liver injury (DILI) [62]. The application of PK principles can significantly enhance TB treatment outcomes, especially in cases involving multidrug resistance [63]. Despite advancements, significant gaps remain in PK data for anti-TB drugs, notably among pediatric patients with TBM [61].

TBM represents one of the most severe extrapulmonary manifestations of TB, necessitating a nuanced approach to treatment that hinges on the effective penetration of anti-TB drugs into the CNS. Among the arsenal of first-line anti-TB drugs, INH and RIF stand out for their critical roles in TBM treatment, albeit with contrasting PK profiles that influence their efficacy and therapeutic strategies.

An essential element of TB management is the ability of drugs to cross the BBB, typically facilitated by lipophilic drugs with a molecular weight under 400 g/mol and fewer than eight hydrogen bonds, allowing lipid-mediated free diffusion across the BBB [64]. However, the penetrative efficacy of anti-TB medications in the CNS varies significantly. RIF, for instance, whether administered orally or intravenously, often results in suboptimal CSF concentrations due to limited CNS penetration [61]. In contrast, INH effectively penetrates the CNS, achieving peak CSF concentrations comparable to plasma levels in both children and adults [65,66,67]. This variation in drug penetration underlines the critical need for dose optimization strategies, particularly for the treatment of extrapulmonary TB (EPTB) manifestations, such as meningitis and bone/joint disease, to ensure therapeutic efficacy and improve patient outcomes. A review analyzing CSF concentrations of INH after administration revealed significant variability across different dosages and methodologies, with INH concentrations in CSF ranging from 0.55 to 14.1 μg/mL after 3 h of administration at a dosage of 1.5 to 20 mg/kg in both children and adults [67]. This variability underscores the complexity of achieving optimal therapeutic levels in the treatment of TBM.

The PK-PD metrics crucial for evaluating the efficacy of anti-TB are the ratio of C_max_ to MIC (minimal inhibitory concentration) and the ratio of AUC (area under curve) at the end of the dosing interval relative to MIC (AUC_0–24_/MIC). These measures highlight the critical role of MIC values in determining drug efficacy against *Mycobacterium tuberculosis* [63]. In a comprehensive study, MICs for 14 anti-TB drugs were assessed against *Mycobacterium tuberculosis* strains. Remarkably, MICs for these drugs consistently remained within a narrow range, indicating susceptibility across all tested strains. The MIC values, reported in µg/mL, demonstrated variability across the drugs: INH (0.02 to 0.04), RIF (0.2 to 0.4), ETB and streptomycin (0.5–2.0), ethionamide (0.25–0.5), D-cycloserine (25–75), capreomycin (CM) (1–2), kanamycin (KAN) (2–4), amikacin (AMK) (0.5–1.0), clofazimine (0.1–0.4), ofloxacin (0.5–1.0), ciprofloxacin (0.25–1.0), and sparfloxacin (0.1–0.4) [68]. Intriguingly, the study revealed that RIF, even at a minimal concentration of 0.2 µg/mL, was effective in inhibiting the metabolism of susceptible bacterial strains. Conversely, at concentrations up to 32 µg/mL, RIF showed minimal impact on drug-resistant strains of *M. tuberculosis*, highlighting the challenge of combating resistance and the necessity for precise dosing strategies [68].

A prospective observational study conducted in Indonesia involving 20 TBM patients provided insights into the PK of INH, RIF, and PZA. This study uncovered that RIF displayed suboptimal RIF AUC_0–24_ levels for TBM treatment and notably low concentrations in CSF. Additionally, a significant correlation was identified between elevated AUC_0–24_ and C_max_ values for INH, RIF, and PZA and the incidence of DILI, emphasizing the need for meticulous drug monitoring. Moreover, within the initial four weeks, four patients encountered grade 2–3 DILI, leading to a temporary discontinuation in drug therapy. However, upon the drugs’ reintroduction, no subsequent cases of DILI were noted. Notably, patients who developed DILI exhibited higher AUC_0–24_ for INH, RIF, and PZA, as well as elevated C_max_ values for INH and PZA on day 10 relative to those who did not experience DILI (*p* < 0.05), indicating a potential link between drug exposure levels and the risk of DILI [61].

RIF encounters significant barriers in achieving therapeutic CSF levels for TBM treatment due to its high molecular weight, impeding its BBB passage. The effectiveness of RIF in treating TBM critically depends on exceeding specific MIC levels in the CSF, a criterion not met by current treatment protocols. This often results in sub-therapeutic drug concentrations in the majority of TBM cases, underscoring the imperative for precise dosage adjustments to amplify RIF’s therapeutic effects. An in-depth examination of RIF’s PKs, including its absorption, metabolism, and excretion, alongside a comparative analysis of oral versus IV administration, reveals the complexity involved in optimizing CNS drug delivery. These insights strongly advocate for customized dosing strategies to enhance TBM treatment efficacy, highlighting the challenges and proposing potential solutions for attaining therapeutic drug concentrations within the CSF.

In a PKs assessment involving 20 participants, INH demonstrated an AUC_0–24_ of 18.5 (5.1–47.4) h∙mg/L, with a C_max_ of 4.6 (1.0–10.0) mg/L. Concentration in CSF varied across time intervals, ranging from 1.4 (0.5–6.1) to 1.6 (1.2–2.5) mg/L in the first two hours, and from 1.3 (1.2–4.3) to 2.3 (1.9–2.8) mg/L between six to eight hours. RIF exhibited an AUC_0–24_ of 66.9 (21.7–118.6) h∙mg/L and a plasma C_max_ of 9.4 (2.9–23.7) mg/L. However, its concentration in CSF was notably lower, ranging from 0.2 (0.1–0.4) to 0.4 (0.1–1.4) mg/L across different intervals. PZA showed the highest AUC_0–24_ of 315.5 (100.6–599.0) h∙mg/L and a C_max_ of 37.7 (15.9–61.7) mg/L. Concentration in CSF ranged from 24.4 (11.1–54.9) to 19.6 (7.2–37.7) mg/L across various intervals. A subsequent evaluation with 12 participants confirmed these patterns, with no significant variances in PK parameters observed between the assessments [61].

#### 3.1.1. PK of RIF in CSF

RIF, a first-line anti-TB medication [69], is notably less effective at reaching therapeutic CSF levels due to its substantial molecular weight (Mw 822.9 g/mol), impeding BBB permeability [70]. Such limitation is exacerbated by plasma protein binding, with only ~10–20% of RIF reaching the CSF [71]. Crucially, for RIF to be effective in TBM treatment, CSF MIC values must exceed 15 µg/mL [22], a condition unmet by existing treatment guidelines, leading to sub-therapeutic RIF levels in up to 89% of TBM patients [72]. This dilemma highlights the necessity of carefully tailored dosage adjustments. RIF shows a dose-dependent elevation in serum concentration [73,74,75,76,77], underscoring the importance of meticulous dosage adjustments to enhance its efficacy. In the treatment of bacterial meningitis, it is crucial to rapidly eliminate bacteria from the CSF to ensure survival without neurological impairment [78]. Achieving therapeutic levels of antimicrobial drugs within the CSF is essential for eradicating bacteria in TBM, influenced by both drug PKs and the strain’s susceptibility [79].

RIF metabolism occurs in the liver, with hepatic esterase transforming RIF into deacetyl-RIF [80]. Both RIF and its metabolite undergo biliary excretion [80,81], with approximately 17% of a 600 mg RIF dose recovered unchanged in urine [81]. Liver excretion saturates between 300–450 mg doses, leading to higher serum concentrations. With repeated doses, the body’s self-metabolism increases during the initial treatment phase, causing serum concentration and half-life to decrease [82]. Roughly 80% of RIF binds to albumin, dispersing across tissues [82]. The elimination half-life of RIF in CSF is extended compared to serum, ranging from 9.1 to 21 h with “uninflamed” meninges, with a median of 14.5 h, *n* = 5). This is notably longer than the serum half-life of RIF, which ranges from 2.2 to 5.8 h, with a median of 3.6 h; (*n* = 7) [83]. The distribution of RIF within the CNS is restricted, impacting its efficacy. In a study using a standard dosing regimen, a large proportion of TBM patients exhibit undetectable levels of RIF in their CSF [84], rarely exceeding 1 μg/mL [67,77,85] as approximately only about ~5% of plasma RIF reaches the CSF due to high protein binding, indicating poor penetration into the CNS, falling below the MIC required for the eradication of *Mycobacterium tuberculosis* and potentially leading to suboptimal dosing and treatment failure [72].

A study examining CSF concentrations of RIF reported variability at different time intervals after administration, with doses ranging from 3 mg/kg to 20 mg/kg in both children and adults. Notably, CSF RIF levels varied from 0.14 to 1.7 μg/mL three hours after administration, indicating the challenges of achieving therapeutic CSF levels with oral RIF in TBM treatment [67]. Such PK insights necessitate the development of optimized dosing regimens, potentially necessitating higher doses for children and adults alike to effectively combat TBM, reduce mortality rates, and address the critical challenges in CNS drug delivery.

This comprehensive analysis underscores the complexity of achieving adequate therapeutic levels of RIF within the CSF for TBM treatment and the urgent need for innovative dosing strategies to improve patient outcomes.

Upon oral administration, RIF is rapidly and completely absorbed in the intestines, especially when taken on an empty stomach. A single 600 mg dose achieves a peak serum concentration of 10 μg/mL within 2 h, corresponding to a half-life of 2.5 h [82]. RIF is metabolized in the liver through deacetylation by hepatic esterase, forming diacetyl-RIF, a more polar and microbiologically active metabolite. It is excreted in similar quantities in both bile and urine, though the presence of diacetyl-RIF is more likely to be found in bile, at a ratio four times higher than its presence in urine [82]. Given its potential for DILI with long-term or high-dose usage, cautious application of RIF is advised for patients with renal and hepatic impairment [86].

A comparative study of oral and IV RIF administration indicated that higher oral doses (750 mg or 16.7 mg/kg and 900 mg or 18 mg/kg) achieve similar total plasma RIF levels as a 600 mg (13.3 mg/kg) IV dose within 1.5 h. IV administration, which has been associated with reduced mortality in TBM, results in higher peak plasma concentrations [85]. In African patients with HIV-associated TBM, the plasma RIF AUC_0–24_ was greater for an oral dose of 35 mg/kg than for an IV dose of 20 mg/kg, though C_max_ values remained comparable [87].

A review of 18 studies revealed that only seven reported RIF CSF concentrations exceeding 1.0 μg/mL [67]. Notably, higher CSF concentrations of RIF were observed in four cases where children received dosages ranging from 15 to 20 mg/kg. Moreover, one study documented mean RIF CSF concentrations of 2.01 μg/mL (range 0.8–3.5 μg/mL) six hours post-dosing in adults with TBM. Another report on seven adults requiring ventriculostomy indicated median ventricular CSF RIF concentrations of 0.73 μg/mL (range 0.57–1.24 μg/mL). Generally, recorded RIF CSF concentrations did not consistently surpass the MIC for *M. tuberculosis*, suggesting suboptimal therapeutic efficacy.

Although some efficacy might persist at these low concentrations, the situation cannot be considered optimal. The highest CSF concentration of RIF is typically observed in children receiving a 20 mg/kg dosage of RIF. Given that children generally exhibit lower serum concentrations of RIF compared to adults at standard dosages (8–12 mg/kg), advocating higher dosages for TBM management in children seems reasonable. Dosages of at least 20 mg/kg in young children and infants weighing less than 10 kg, and at least 15 mg/kg in those weighing between 10 and 20 kg, are suggested. The optimal CSF concentration of RIF was typically seen in children at a 20 mg/kg dosage, suggesting a need for higher dosage recommendations for TBM management in pediatric populations [67].

Ellard and colleagues emphasize the vital importance of RIF in the treatment of TBM, despite its limited penetration into the CSF. The significance of RIF, especially in addressing drug-resistant TB strains and managing severe disease symptoms, including widespread lesions and the need for an effective therapeutic response, is highlighted. This focus emphasizes the necessity for comprehensive treatment strategies, particularly in the face of INH resistance, to ensure an effective response against TB. This emphasis sheds light on the necessity for robust treatment approaches, particularly in scenarios where INH resistance is present, to secure an effective therapeutic response against TB [88].

In a study involving 27 Chinese TBM patients, the PK of RIF following oral dosage (11 mg/kg) revealed slow penetration of RIF concentrations in serum and CSF from 2 to 6 h. Serum concentrations were initiated at 11.5 mg/L (2 h) and declined to 4.7 mg/L (6 h), maintaining levels slightly above the MIC for *M. tuberculosis* (0.3 mg/L). Poor CSF penetration was observed, with CSF/serum ratios increasing from 0.04 (2 h) to 0.11 (6 h) [88]. This highlights challenges in achieving therapeutic CSF levels with oral RIF in TBM treatment. PK targets, such as RIF AUC_24_ (at least 116 μg·h/mL (equivalent to AUC6 of 70 μg·h/mL) and C_max_ (22 μg/mL) have been recommended for optimizing TBM treatment, emphasizing the need for adequate drug exposure in the CSF [89]. Peak CSF concentrations, observed 0–8 h (median = 1 h) post-infusion completion, varied between 0.57 to 1.24 mg/L (median = 0.73 mg/L) [83]. PK modeling studies indicate that for childhood TBM, RIF doses of at least 30 mg/kg orally or 15 mg/kg intravenously may be necessary to attain adequate CSF drug concentrations and effectively eradicate bacteria, thus lowering mortality rates. However, it remains unclear whether similar dosing regimens are applicable to adults [90].

IV administration of RIF has shown promise, achieving higher plasma and CSF concentrations compared to oral dosing, potentially offering a pathway to optimizing CNS drug delivery. IV RIF demonstrates higher plasma and CSF concentrations compared to oral dosing, suggesting potential benefits for optimizing CNS drug delivery [84]. Addressing drug penetration challenges and optimizing dosing strategies are crucial for improving TBM treatment outcomes.

Despite high-dose IV RIF administration, mortality rates in TBM remain high, with a 35% mortality rate (at 13 mg/kg), with a comparatively higher rate of 65% observed in standard doses reported by Ruslami et al. [84]. In a study of 26 patients receiving IV (600 mg) and oral (450 mg) doses, IV administration significantly increased plasma AUC_0–6_ (78.7 mg.h/L) compared to oral (26.0 mg.h/L), with a ratio of 3.0. IV C_max_ in plasma was markedly higher (22.1 mg/L) than oral (6.3 mg/L), with a ratio of 3.5. Median T_max_ for both routes was 2 h, but IV administration showed a significantly narrower range (1–2 h) compared to oral (1–6 h, *p* = 0.048). In CSF, IV C_max_ (0.60 mg/L) was significantly higher than oral (0.21 mg/L), with a ratio of 2.92 (*p* < 0.0001) [84]. Table 1 details the systemic therapeutic regimen, CSF penetration and side effects of anti-TB in the treatment of TBM.

In summary, clinical observations and PK modeling studies suggest that higher doses of RIF may be necessary to attain adequate CSF drug concentrations, a strategy that might help reduce mortality rates in TBM. However, the risk of DILI and the PKs of RIF, including its metabolism and excretion, demand careful consideration.

#### 3.1.2. PK of INH in CSF

INH, a hydrophilic antimicrobial, exemplifies the potential for effective BBB penetration despite the challenges associated with delivering drugs to the CNS. Unlike lipophilic agents that cross the BBB via lipid-mediated diffusion, INH’s water solubility enables it to traverse the BBB paracellularly, achieving significant concentrations within the CSF. This property is critical for combating *Mycobacterium tuberculosis* within a compartment notoriously difficult to reach. Clinical PK analysis demonstrates that INH attains steady-state concentrations in both plasma and CSF, with peak levels occurring shortly after administration and maintaining bactericidal activity within the CNS. The WHO’s recommendation for higher INH dosages in the treatment of MDR/XDR-TBM/PTB acknowledges this drug’s capacity for high CNS penetration and underlines the importance of achieving therapeutic CSF concentrations to combat resistant TB strains effectively.

The PKs of INH, another important first-line TB drug, in CSF are important for the effective treatment of TBM [93]. Unlike lipophilic agents, hydrophilic antimicrobials like INH and PZA cross the BBB paracellularly due to their water solubility [93]. Optimal BBB penetration occurs when drug log *p*-values range from 1.5 to 2.7, with a mean value of 2.1 [94]. INH, a hydrophilic drug (Mw 137.14 g/mol), freely penetrates the BBB (80–90% into CSF), exhibiting potent bactericidal activity [95,96,97]. CSF penetration is predicted by the AUC_csf_/AUC_serum_ ratio, with INH showing an AUC_CSF_/AUC_Serum_ close to 1.0, making it valuable for treating CNS infections [36,98]. The MIC of INH in liquid media ranges from 0.02 to 0.04 µg/mL [68]. In confirmed TBM patients, a PK analysis of INH transfer into CSF demonstrated steady-state concentrations in both plasma and CSF. Peak plasma levels (4.17–21.5 µg/mL) occurred 0.25 to 3 h post a multiple INH dose (600 mg/day). Parameters included a terminal half-life of 1.42 ± 0.41 h, total clearance (CI/F) of 0.47 ± 0.22 L/kg/h, and volume of distribution (Vd/F) of 0.93 ± 0.48 L/kg. In CSF, INH concentrations were highest at 3 h (Mean, 4.18 μg/mL) and 0.54 ± 0.21 µg/mL at 12 h post the last dose of INH 10 mg/kg/day. Using a modified PK/PD model, the disposition rate constant from CSF-to-plasma and CSF/plasma partitioning ratio of INH was estimated to be 0.39–1 h and 1.17, respectively [99]. Similarly, in a study involving 27 Chinese TBM patients, INH exhibited rapid diffusion into the CSF after oral administration of INH (9 mg/kg) [88] and attained peak concentrations surpassing 3 mg/L, exceeding its MIC against *Mycobacterium tuberculosis* by over 30 times, achieved within 4 h [28]. Mean serum concentrations peaked at 4.4 mg/L within 2 h, decreasing to about 1 mg/L by 6 h. CSF concentrations reached 1.9 mg/L within 2 h, and increased to 3.2 mg/L by 4 h, surpassing 30 times its MIC against *Mycobacterium tuberculosis*. CSF/serum INH ratios rose from approximately 0.5 at 2 h to 2.1 at 6 h, unaffected by steroid administration [88].

Considering the good absorption and penetration of INH in CSF, achieving therapeutic concentrations in CSF is feasible. However, to prevent systemic side effects, IT administration of INH is preferred. Typically, a 24 h dosing interval is sufficient for moderately lipophilic antibiotics (MW > 1000 g/mole) and hydrophilic drugs (MW > 400 g/mole). Therapeutic drug monitoring (TDM) is necessary only in cases of treatment failure [40].

For the treatment of MDR/XDR-TBM/PTB, the WHO recommends 15–20 mg/kg INH, recognized for its high CNS penetration [65,66]. Utilizing a high dose (800 mg/d), peak CSF concentration (11.57 μg/mL) was observed after 6 h, indicating nearly 100% brain penetration and effective bactericidal activity [61]. Long-term use of INH may lead to peripheral neuropathy at the sixth month, with reported neuritis [92]. The MIC of RIF for pre-XDR-TBM found in a case study > 16 µg/mL and for INH is 0.06 µg/mL while the reference value is <1 and <0.25, respectively [92]. 

In summary, the ability of INH to penetrate the BBB and reach concentrations above the MIC underscores its potential effectiveness in treating TBM. High doses of INH can enhance BBB penetration, leading to higher concentrations in the CSF. However, prolonged use of INH is linked to side effects such as peripheral neuropathy and neuritis.

#### 3.1.3. Comparative PK with Other Anti-TB Drugs in CSF

The treatment of TBM, particularly MDR forms, necessitates a nuanced understanding of the PK of various anti-TB drugs to ensure effective CNS penetration and therapeutic outcomes. This editorial examines the PK profiles of linezolid, bedaquiline, and clofazimine, comparing their efficacy and challenges in treating TBM.

Linezolid: a promising option for MDR-TB. Linezolid, an antibiotic effective against MDR-TB, demonstrates significant promise due to its favorable PK profile in both serum and CSF. Administered 600 mg twice daily via IV infusion, linezolid achieves a mean C_max_ of 18.6 μg/mL in serum and 10.8 μg/mL in CSF, with a CSF-to-serum penetration ratio of 0.66. Notably, linezolid’s elimination half-life in the CSF (19.1 h) surpasses that in serum (6.5 h), suggesting sustained activity within the CNS [100]. This extended half-life in the CSF is particularly advantageous for treating TBM, allowing for more consistent antimicrobial pressure against *Mycobacterium tuberculosis*.

Bedaquiline: challenges and potential. Bedaquiline, another critical component in the treatment of MDR-TB, presents a more complex PK profile. While plasma concentrations fall within expected ranges, indicating adequate systemic absorption, CSF penetration appears limited. Bedaquiline’s C_max_ in the CSF is significantly lower than in plasma, reflecting a CSF-to-plasma concentration ratio of merely 0.12%. PKs were studied with thirty-eight plasma and seven CSF samples. Plasma concentrations were within the expected range, with a bedaquiline C_max_ of 1368.1 ng/mL and AUC last of 19,825.9 ng·h/mL; M2 reached a C_max_ of 217.3 ng/mL and AUC last of 4134.6 ng·h/mL. In CSF, bedaquiline had a C_max_ of 3.790 ng/mL, while M_2_ had a C_max_ of 1.400 ng/mL. CSF-to-plasma concentration ratios were 0.12% for bedaquiline and 0.3% for M_2_, aligning with estimated plasma unbound fractions [101]. Despite this, the inclusion of bedaquiline in intensified treatment regimens has contributed to improved clinical outcomes in MDR-TBM, suggesting that even limited CNS penetration can be clinically beneficial when combined with other effective anti-TB agents.

Clofazimine: limited CNS penetration. Clofazimine, despite its efficacy in treating MDR-TB, exhibits limited benefits for TBM due to its poor CNS penetration. In a case report, plasma concentrations reached peak levels post-administration, while CSF levels remained undetectable. The plasma concentration peaked at 0.35 μg/mL 6 h post-administration, while CSF levels were undetectable. Despite its lipophilicity, the drug’s high plasma protein binding rate (>85%) may limit CNS penetration. Human efflux transporters could further impede brain and CSF access. Findings suggest limited benefits of clofazimine in treating MDR- or pre-XDR-TBM [92]. The drug’s high plasma protein binding rate and potential interaction with human efflux transporters may impede its CNS access, highlighting the challenges of achieving therapeutic CSF concentrations with certain anti-TB drugs.

Intensified treatment regimens: a path forward. An intensified treatment regimen, including fluoroquinolone, injectables (AMK/KAN/CM), linezolid, carbapenem, RIF, INH, PZA, clofazimine, delamanid, and bedaquiline, administered for 361 days in drug-susceptible TBM and 486 days in MDR-TBM, resulted in a 9% mortality rate in the MDR group [102]. Neurological outcomes improved over time, with 18% reporting a Modified Rankin Scale (MRS) score of 0 at 1 month, increasing to 85% at 6 months, and further to 94% at 12 months. Disability rates also showed improvement, with 36% having low Glasgow Outcome Scale (GOS) disability at 1 month, 80% at 6 months, and 86% at 12 months. These findings underscore the positive impact of intensified regimens on mortality and neurological outcomes in TBM. Increased use of drugs that penetrate well into the CSF, including LZD, may be one reason for our favorable outcomes, given the concern for limited CSF penetration of first-line anti-TB agents (reported by a prospective cohort study conducted in the National Center for TB and Lung Diseases, Georgia) [102].

The success of intensified treatment regimens, incorporating drugs like linezolid and bedaquiline alongside standard anti-TB medications, underscores the importance of optimizing drug selection based on PK profiles and CNS penetration capabilities. Such regimens have shown a significant positive impact on mortality and neurological outcomes in TBM, particularly in MDR cases. The careful combination of drugs with varying CSF penetration efficiencies allows for a comprehensive approach to combating TB within the CNS, addressing the limitations posed by individual drugs.

The comparative PK of linezolid, bedaquiline, and clofazimine highlights the complexities of treating TBM, especially in the context of drug resistance. While challenges such as limited CNS penetration persist, the strategic use of these drugs within intensified treatment regimens offers hope for improved outcomes. Future research should continue to focus on understanding the PK properties of anti-TB drugs, exploring new delivery methods, and developing regimens that maximize CNS penetration and therapeutic efficacy against TBM. In summary, the PK of anti-TB drugs is critical for treatment success, with challenges like limited CNS penetration. Key drugs, such as RIF, face difficulties in reaching therapeutic levels in CSF. However, INH demonstrates better CNS penetration. An intensified treatment regimen, administered long-term (361 days for drug-susceptible TBM and 486 days for MDR-TBM), yields improved clinical outcomes. Linezolid, with enhanced CSF penetration, holds promise for better results, addressing concerns about limited penetration of first-line anti-TB agents. IT/IVT anti-TB treatments show superior efficacy with short durations and minimal side effects. Table 2 presents the PK indices of the anti-TB treatments.

### 3.2. Clinical Experience and Evidence Supports Concomitant IT/IVT Anti-TB Effectiveness and Safety

Administering drugs directly into the spinal canal offers benefits, including precise targeting of the CNS, reduced overall doses, and improved treatment outcomes. However, the absence of standardized treatment regimens underscores the necessity of leveraging existing data to develop consistent drug protocols. Analysis of published literature, encompassing case reports, cohort studies, meta-analyses, and other documents, has yielded compelling evidence supporting the effective clinical use of RIF and INH via IT/IVT administration. Our literature review examined the use of RIF, INH, rifabutin, and fluoroquinolones administered via IT or IVT for treating TBM and meningitis caused by *Flavobacterium meningosepticum*. Extensive documentation on INH and RIF, dating back to 1955 and 1976, respectively [105,106], reveals promising outcomes even in cases of refractory TBM, without serious side effects. Despite limited PK data, evidence supports the efficacy of IT/IVT administration in managing TBM.

Analyzing the published literature, including case reports, cohort studies, meta-analyses, and additional literature pieces, we extracted compelling evidence supporting the effective clinical use of RIF and INH through IT/IVT administration.

IT/IVT INH: Historically, in 1955, the treatment of twelve cases of TBM with IT-INH doses ranging from 10 to 20 mg per day led to significant clinical improvement in ten patients. Follow-up assessments, conducted over a period of up to 14 months, reported no toxicity or side effects associated with this treatment regimen [105]. In another instance, 18 patients treated with IT-INH demonstrated the infection eradicated in 17 cases, but two deaths occurred. Patients received streptomycin and oral INH, with doses ranging from 25 to 75 mg by weight for streptomycin and 5–7 mg/lb daily for oral INH, with IT doses of 25–50 mg. Most patients fully recovered, some with neurological improvement. IT-INH injection at approximately 0.5 mg/lb was safe and well-tolerated, but doses below 0.3 mg/lb showed a slower treatment response. Side effects were minimal, including non-specific meningeal irritation in one infant and major epileptic attacks in an adult with pre-existing conditions. This approach, combined with systemic therapy, was effective in reducing the need for frequent streptomycin injections, indicating most patients recovered fully or showed significant neurological improvement [107].

Similarly, a patient developing hepatitis from high oral doses of INH (1000 mg/day) showed clinical improvement upon gradual reintroduction of IT and oral INH. Consequently, INH was discontinued and later re-administered in gradually increasing IT and subsequently oral doses, up to the final dose of 400 mg/day. The patient was treated with IT-INH at a dosage of 35–37 mg every 3 days, resulting in an improvement of clinical symptoms [108]. A study involving 23 TBM cases demonstrated successful treatment with 100 mg of INH and 2 mg dexamethasone administered directly into the lateral ventricle, showing positive outcomes with IT administration [24]. A case of refractory TBM treated with IT-INH at 100 mg three times per week demonstrated significant symptom improvement without any observed toxicity. Following the commencement of IT therapy, there was an immediate improvement in the patient’s consciousness disturbance and CSF findings. One month later, a ventriculoperitoneal shunt operation was performed to address hydrocephalus, leading to further clinical improvement [109]. In another study, 12 pediatric cases with severe TBM, hydrocephalus, and altered consciousness received IT INH and dexamethasone injections. From day 2 onwards, doses of 5 to 20 mg of INH and 1 mg of dexamethasone in 5 mL of saline were administered into the ventricle through the Ommaya reservoir using a syringe, with injections repeated once every 2 to 3 days. Successful recovery was observed in nine out of twelve cases [110]. A 30-year-old Vietnamese woman, 19 weeks pregnant, presented with acute cerebral infarction, left middle cerebral artery stenosis, TBM, and miliary tuberculosis. Treatment with heparin, quadruple anti-TB therapy, and dexamethasone led to rapid symptomatic improvement, but she experienced a stillbirth, followed by recurrent acute cerebral infarction with LMCA occlusion, sinus thrombosis, and cranial base inflammation. Thrice-weekly 100 mg IT-INH improved meningeal inflammation signs. Discharged ambulatory after 7 months, suggesting multimodal therapy with IT-INH and steroids for refractory TBM, as reported by Nakatani et al. [111]. In treating 10 ventricular TB patients with IT-INH (0.1 g) and dexamethasone (2.5–5 mg) alongside multidrug anti-TB therapy, patients showed positive clinical outcomes. MRI findings demonstrated complete resolution or significant reduction in the size of tubercular lesions in seven out of eight patients. These results suggest the potential effectiveness of combining IT INH and corticosteroids for intracranial TB treatment [112].

The effectiveness of INH is significantly influenced by the activity of the N-acetyltransferase 2 (NAT2) enzyme in the liver, which is categorized into slow, intermediate, or rapid acetylation rates. These rates affect the metabolism of the drug, thereby impacting its efficacy in treatment. The presence of variations in NAT2 genotypes necessitates the adjustment of INH dosages to optimize treatment outcomes, particularly in populations with a high prevalence of rapid acetylators. Rapid acetylators metabolize INH more quickly, leading to lower concentrations of the drug in the plasma and an increased risk of treatment failure [113,114]. Given the notable prevalence of rapid acetylators in the Chinese population, higher doses of INH might be essential to improve treatment outcomes [115]. Additionally, recent studies suggest the potential benefits of administering high doses of INH for managing cases of MDR-TB [116,117]. Administering high doses of INH, especially through IT routes, holds promise for treating MDR-TB. This approach can potentially increase the concentration of the drug in the CSF while minimizing the adverse effects associated with higher systemic doses [108,118].

IT/IVT-RIF: In meningitis caused by *Flavobacterium meningosepticum*, RIF was administered intravenously and directly into the cerebral ventricles for neonatal cases. IVT dosage varied from 2 to 5 mg/day for 10 days in two cases and for 10 weeks in one case, in conjunction with systemic therapy. This approach led to rapid bacterial clearance from the CSF. Notably, jaundice was observed as a side effect, and the majority of infants developed hydrocephalus, necessitating shunt placement. Nevertheless, two infants achieved normal neurological development following treatment [106]. In a review of treatment strategies from 1977, a series of seven meningitis cases attributed to the same bacterium were managed with IVT-RIF, dosed between 2 to 5 mg/day until achieving CSF sterilization. Treatment was complemented with intramuscular RIF at 20 mg/kg every 12 h. Transient jaundice was a noted side effect, with ventricular antibiotic concentrations evaluated 20 to 24 h post-administration [119]. Further reports highlighted a treatment regimen consisting of concurrent intravenous RIF at 40 mg/kg per day and IVT-RIF at 5 mg daily for 22 days, resulting in CSF sterilization without reported adverse effects or toxicity [120]. An evaluation of nine meningitis cases treated with either IT or IVT-RIF at 3 mg/kg/day for a minimum of 10 days showed that five patients achieved CSF sterilization and survival. While two deaths were due to other causes, two other patients did not survive the infection. The combined use of intravenous RIF, moxalactam, and piperacillin emerged as a promising antibiotic regimen for treating patients with normal or mildly dilated ventricles. In cases where significant ventriculomegaly was present, the simultaneous intravenous and IVT administration of antibiotics, tailored to the organism’s sensitivity, proved necessary for eradicating the infection. Remarkably, no side effects or toxicity were reported throughout these treatment courses [121]. Additionally, two infections treated with IVT-RIF at a daily dose of 5 mg for 7–10 days were documented, successfully leading to CSF sterilization without any reported side effects or toxicity [122]. These meningitis patients were also treated with a systemic regimen of clindamycin, RIF, and cefotaxime, complemented by IVT administration of RIF.

A case of TBM involved a patient experiencing complications and a relapse after oral anti-TB treatment, who was then treated with IVT-RIF. The treatment regimen comprised an initial daily dose of 5 mg RIF for one week, followed by 3 mg every other day for three additional weeks. Despite a brief interruption due to technical issues and patient non-compliance, treatment resumed after drain placement and continued for another month, aiming to achieve a CSF concentration exceeding 15 μg/mL. This integrated approach successfully effectively managed the meningeal infection, leading to significant clinical improvement without adverse effects [22]. IVT-RIF was highly effective in severe cases of TB meningoencephalitis. In one instance, a patient with concurrent hepatic and renal failure underwent successful treatment with daily 5 mg IVT-RIF administered via an Ommaya reservoir. The 50-day course of IVT-RIF treatment was well-tolerated and demonstrated high efficacy in conjunction with systemic therapy [23]. Furthermore, an accidental high-dose administration of IT-RIF at 600 mg in a postoperative spine infection case showed no immediate or 6-month adverse effects, indicating the potential safety margin of the drug in acute settings [123].

IT/IVT fluoroquinolones in the treatment of MDR-TBM: The application of IT fluoroquinolones has been observed in the treatment of TBM. Notable, a case of MDR-TBM was effectively managed using a flexible regimen. The regimen comprised alternating daily IT administration of LVX at a dose of 1.5 mg and AMK at 5 mg with IV doses of LVX at 500 mg and AMK at 1200 mg. This was in conjunction with oral anti-TB medications and prednisolone at a dosage of 25 mg. Adjustments to the regimen were made to accommodate the patient’s tolerance and the CSF concentration, leading to positive clinical and microbiological results. The targeted CSF concentrations (LVX: 8–10 µg/mL; AMK: 40 µg/mL) were successfully achieved [25].

In summary, the evaluation of IT/IVT anti-TB (RIF, INH, and fluoroquinolones) in managing TBM yields promising outcomes, supported by existing data demonstrating their efficacy and safety, particularly in clinical cases and even in refractory TBM. The summarized literature under the subtitle “clinical experiences and evidence supports IT/IVT anti-TB effectiveness and safety” is provided in Table 3, affirming the high effectiveness and safety of IT/IVT anti-TB drugs.

### 3.3. Optimization of IT Therapeutic Regimens of RIF and INH in the Treatment of TBM

Despite the absence of an optimized IT/IVT therapeutic regimen for anti-TB treatment in TBM, our analysis of published data aims to propose a regimen to enhance treatment strategies, ultimately improving patient outcomes and reducing associated morbidity and mortality. We have synthesized data from case reports, cohort studies, and literature articles to optimize therapeutic regimens for IT/IVT administration of anti-TB drugs in TBM management.

RIF: In seven published literature, IT/IVT-RIF was given at doses of 2–5 mg for 7–50 days in seven studies. This regimen proved highly effective, especially in complex cases and when oral drugs failed, with no significant side effects [22,23,106,120,121,122,123]. The targeted concentration of RIF in the CSF was 15 mg/L. See Table 3 for dosage and treatment duration details [22].

INH: Similarly, IT/IVT administration of INH, typically given thrice weekly at dosages between 5 to 100, proves highly effective with a mild side effect. Details of dosage and treatment duration are provided in Table 3 [24,105,107,108,109,110,111,112].

LVX and AMK: The recommended dosage for IT fluoroquinolones (LVX and AMK) in treating TBM is LVX 1.5 mg and Am 5 mg, supplemented with IV-LVX 500 mg and Am 1200 mg, oral anti-TB drugs, and 25 mg prednisolone. This flexible regimen, tailored to patient tolerance and CSF concentration, has yielded positive clinical and microbiological outcomes, achieving target CSF levels (LVX, 8–10 g/mL; AMK, 40 μg/mL) [25].

Figure 1 depicts the mean and median doses of IT/IVT-RIF, INH, and fluoroquinolones for TBM treatment, sourced from the published literature.

The synthesis of data from various sources informs optimized therapeutic regimens for IT/IVT administration of anti-TB drugs in TBM management, emphasizing efficacy and minimal side effects.

### 3.4. Comparative Analysis of Systemic Treatment versus Concomitant IT-Anti-TB Therapy

A comprehensive assessment comparing systemic anti-TB therapy to concomitant IT-Anti-TB therapy is essential. However, there is a notable lack of studies that thoroughly compare these approaches across all anti-TB medications in the treatment of TBM. In evaluating combined systemic and IT/IVT dosing, understanding the central volume of distribution is critical, as it correlates with CSF concentrations in both compartments, potentially influencing their effectiveness [52].

Comparing IT-INH alongside systemic therapy vs. systemic INH alone: A propensity-matched cohort study compared IT-INH alongside systemic therapy to systemic anti-TB therapy alone, highlighting the advantages of the concomitant treatment approach. This study showed that integrating IT therapy with systemic anti-TB therapy was more effective than systemic therapy alone in patients with TBM over long-term follow-up [124]. In the overall cohort, there were 36 patients in the IT group and 162 patients in the systemic therapy group. Among the propensity-matched cohort comprising 30 patients in each group, the analysis revealed noteworthy findings. Specifically, the odds ratio (OR) for death in the propensity-matched cohort was 0.266 (95% CI: 0.073–0.964; *p* = 0.037), indicating a significantly lower risk of mortality in the IT therapy group compared to the systemic therapy alone group. Furthermore, the OR for poor outcomes was 0.386 (95% CI: 0.136–1.094; *p* = 0.073), suggesting a trend towards improved outcomes in the IT therapy group, although statistical significance was not reached. These findings underscore the potential benefit of incorporating IT therapy with systemic anti-TB in the management of TBM, particularly in reducing mortality rates [124].

Table 3 outlines the doses, regimens, and efficacy of RIF, INH, LVX, and AMK when administered via IT and IVT administration routes for the treatment of TBM. The data spans from 1955 to the present and encompasses all relevant literature on IT/IVT anti-TB therapies.

In conclusion, while there is a scarcity of data to directly compare the clinical outcomes of all anti-TB drugs between concomitant IT/IVT therapy and systemic anti-TB drugs alone for treating TBM, clinical observations and case studies, including published case reports and cohort studies on TBM patients, suggest and support the practice of administering IT/IVT alongside systemic anti-TB therapies concurrently. pharmaceutics-16-00540-t003_Table 3Table 3IT/IVT Administration of RIF, INH, LVX, and AMK in Anti-TB Therapy.DrugsDiseasePtsDosage and RegimensEfficacyToxicity/Side EffectReferencesINHTBM1210–20 mg/d IVTCuredConvulsion 1, partial optic atrophy 1Sifontes, J. et al., 1955 [105]TBM1825–50 mg IT for 1~119 daysCuredHemiplegia 2, etc.Swift, P. et al., 1956 [107]TBM135–37 mg IT every 3 daysCuredHepatotoxicityDaielides, I., 1983 [108]TBM23100 mg/d or qod IT based on the severityCured8 pts mild brain hernia rapidly improvedChen, Y. et al., 1996 [24,125]TBM1100 mg, 3x/wk IT.CuredNoTakahashi, T., 2003 [109]TBMH125–20 mg one time every 2–3 days9 cured, 1 Pt died, 2 Pts not cured2 pts disabilityLin, J. et al., 2012 [110],TBM1100 mg, 3x/wk, tapered to 1x/wk upon symptom improvement.CuredMild aphasia, 1 pts hemi-paresisNakatani, Y et al., 2017 [111]TBM10100 mg/d ITCuredOne patient, hemiparesisLi, D. et al., 2017 [112]RIFF.M32–5 mg/d, IVT for 10 days (2 patients) or 10 wks (others).CuredTransient JaundiceLee, E. et al., 1976 [106]F.M72–5 mg/d, IVTCuredTransient JaundiceLee, E. et al., 1977 [119] F.M15 mg 1x/d, IVT/ for 22 daysCuredNoRios, I., 1978 [120]TBM15 mg/d, IVT for 7 days, then 3 mg qod for a total of 63 days.CuredNoDajez, P et al., 1981 [22]F.M93 mg/kg/d, IVT for at least 10 days5 cured, 2 died of other DS, and 2 succumbed.NoBoo, N. et al., 1989 [121]F.M 25 mg/kg/d, IVT for 7–10 daysCuredNoBruun, B. et al., 1989 [122]TBM15 mg/d IVT for 50 daysCuredNoVincken, W et al., 1992 [23]TBM1600 mg infusion 1x for 4 h, ITInadvertent Inj.No Senbaga, N et al., 2005 [123]LVX/AMKMDR-TBM11.5 mg LVX and 5 mg AMK qod/ITCuredMild; Insomnia and MyalgiaBerning, S.E. et al., 2001 [25]Note: Ds, disease; Pts, number of patients; MIC, minimum inhibitory concentration; RIF, rifampicin; INH, isoniazid; LVX, levofloxacin; AMK, amikacin; F.M, *flavobacterium meningosepticum*; TBM, tuberculosis meningitis; /d, per day; qod, every other day, wk, week.

### 3.5. Adverse Events and Risks of IT/IVT Administration of Anti-TB Drugs

Similar to other clinical procedures, the IT/IVT administration of anti-TB drugs, while minimally invasive, is not without risks and adverse effects. Data from various case reports and studies indicate the use of IT/IVT-RIF for *Flavobacterium meningosepticum* and *Mycobacterium tuberculosis* in TBM treatment, typically administered at 2 to 5 mg/day over 7 to 50 days. Reversible jaundice is reported in some cases, demonstrated in Table 3 [22,23,106,120,121,122,123]. Notably, even an accidental IT infusion of 600 mg RIF over 4 h showed no adverse events [123]. IT/IVT administration of INH, given thrice weekly at dosages between 5 to 100 mg, is associated with side effects like hemiplegia, quadriplegia, convulsion, partial optic atrophy, and hydrocephalus. Caution is advised based on reported cases, as demonstrated in Table 3 [24,105,107,108,109,110,111,112]. The total side effects and risks of IT/IVT INH and RIF in our review include convulsions, partial optic atrophy, hemiplegia, hepatotoxicity, mild brain hernia that rapidly improved in one patient, mild aphasia, transient jaundice, insomnia, and myalgia, and no significant side effects in most patients noted, as demonstrated in Table 3.

## 4. Discussion

This narrative review indicates that most anti-TB drugs, specifically RIF and clofazimine, exhibit poor CSF penetration when administered systemically, leading to suboptimal concentrations and ineffective therapeutic outcomes. To overcome these issues, healthcare professionals are considering the IT and IVT administration of INH and RIF. This approach is inspired by the successful use of IT analgesics [126], and the treatment of Gram-negative bacterial ventriculitis with IVT antibiotics, including RIF [119].

### 4.1. PK Properties of Anti-TB Drugs

The PK properties of anti-TB drugs within the CSF are pivotal in determining the efficacy of treating TBM and optimizing dosage. This review article addresses the challenges linked with conventional treatments for CNS infections, with a specific focus on TBM. It explores the potential of IT administration of RIF and INH as a targeted drug delivery strategy to mitigate systemic exposure, bypass the BBB, achieve elevated or adequate concentrations of anti-TB drugs in the CSF surpassing the MIC for a sufficient duration, and minimize side effects. A main obstacle identified is the lack of adequate PK data for the direct IT/IVT administration of these drugs. Nonetheless, the findings underscore the efficacy of IT/IVT administration of RIF and INH, demonstrating favorable clinical outcomes without serious side effects, utilizing short durations and small doses. Physiological and PK factors influencing IT anti-TB therapy are dissected, emphasizing the need for therapeutic regimen optimization. Barriers such as the BBB and BCSFB hinder drug entry, although infection-induced meningeal inflammation may enhance penetration for certain antibiotics [36,37].

In this study, we analyzed and summarized the differences in the concentration ratios of drugs in the CSF compared to serum concentrations following oral and IV administration, with a focus on RIF and INH. The comparison between oral and IV administration of RIF revealed distinct differences in drug concentration levels in the CSF. Our analysis, presented in Table 2, showed that oral administration resulted in a mean CSF concentration of 0.49 µg/mL, with a C_max_ of 0.21 mg/L following a 450 mg dose. Conversely, IV administration achieved a higher mean CSF concentration of 0.60 mg/L after a 600 mg dose, indicating more effective penetration into the CSF. These findings indicate that only 5 to 20% of the plasma concentration can reach the CSF through the two different administration routes. However, the evidence suggests that the difference is dose-dependent rather than being solely determined by the route of administration. In contrast, for INH, oral administration resulted in an average CSF concentration of 3.55 µg/mL, ranging from 1.3 to 11.57 µg/mL. The data presented significant differences between RIF and INH regarding their PKs and efficacy against MTB in the treatment of CNS infections. INH showed a much higher average CSF concentration (3.55 µg/mL) compared to RIF (0.49 µg/mL for oral administration), indicating better penetration of INH into the CNS.

The CSF/serum ratio significantly differs between the two drugs, with INH showing a much higher ratio (0.47–2.12) than RIF (0.04–0.11). This suggests that INH is more readily transported from the serum into the CSF compared to RIF. Both drugs exhibited a T_max_ of 2 h post-oral administration for RIF, but INH has a broader range of T_max_ (2 to 4 h), suggesting a slower absorption into the bloodstream in some cases. RIF had a longer half-life (9.1 to 21 h) compared to INH (approximately 1.42 h), indicating that RIF remained in the CSF longer than INH, potentially extending its antimicrobial activity within the CNS.

Regarding MIC values, for RIF, MIC against susceptible MTB ranged from 0.02 to 0.04 µg/mL but was >16 µg/mL for pre-XDR MTB, and varied from >8 to >32 µg/mL for multidrug-resistant (MDR) and rifampicin-resistant (RR) MTB. For INH, MIC against susceptible MTB ranged from 0.02 to 0.04 µg/mL, 0.06 µg/mL for pre-XDR MTB, and from 0.4 µg/mL to >4 μg/mL for MDR- and INH-resistant MTB. RIF and INH both had varying MIC values against different strains of MTB, with INH generally requiring lower concentrations to inhibit susceptible MTB. However, RIF shows efficacy over a broader range of MTB strains, including pre-XDR and MDR/RR MTB, albeit at higher concentrations.

Overall, these differences highlighted the unique PK properties and effectiveness of RIF and INH in treating CNS infections, underscoring the need for tailored therapeutic strategies based on the drug’s ability to penetrate the CNS and its activity against various MTB strains. The findings suggested that IT or IVT administration may offer better treatment outcomes for CNS infections. Nevertheless, the choice of administration route should also take into consideration patient-specific factors, such as condition, drug tolerance, and potential side effects.

### 4.2. The Clinical Significance of Concurrent IT/IVT Therapy

#### 4.2.1. Concurrent IT/IVT Therapy of Anti-TB Drugs

Concurrent therapy and its clinical outcomes of anti-TB drugs: Concurrent IT and IV therapy alongside systemic treatment is crucial for preventing bacterial resistance and reducing relapse risk, especially in challenging cases like MDR-TBM. Simultaneous administration of RIF and INH has shown effective outcomes with minimal side effects, utilizing short durations and small doses. The clinical outcomes of concomitant IT/IVT anti-TB with systemic anti-TB in the treatment of TBM show promise. Clinical evidence supports the efficacy and safety of IT/IVT administration, particularly for RIF and INH, even in refractory TBM cases [22,23,24,105,106,107,108,109,110,111,112,120,121,122,123]. Some adverse events, such as infections, nerve damage, transient jaundice, seizures, etc., have been observed following the IT or IVT administration of RIF and INH [22,23,24,105,106,107,108,109,110,111,112,120,121,122,123]. However, from our perspective, it is challenging to ascertain whether these side effects are directly attributable to RIF and INH or if they stem from disease progression or the specific procedure and route of administration. These events necessitate careful consideration and monitoring. Comparing concurrent administration of IT/IVT anti-TB alongside systemic therapy of anti-TB in the treatment of TBM, there is a lack of data, although one study compared the concomitant IT INH versus systemic alone and showed superior concomitant therapy [124].

#### 4.2.2. Concurrent IT/IVT Therapy of Non-Anti-TB Antibiotics

The IT/IVT administration of non-anti-TB antibiotics such as vancomycin, colistin, tigecycline, and others, plus systemic therapy, has been studied in several papers, and we have documented them here. Concurrent administration of IT/IVT antibiotics alongside systemic treatment is essential in preventing bacterial resistance and reducing the risk of relapse, particularly in cases involving MDR pathogens where systemic options are limited, as acknowledged in CNS infections caused by MDR pathogens and postoperative meningitis or ventriculitis, which highlight the potential life-saving benefits of combining IT/IVT and systemic antimicrobial therapy [40,127]. This strategy allows for achieving high concentrations of antimicrobial agents within the CNS while minimizing systemic side effects, influenced by drug properties and concurrent systemic dosing [56,59,128,129,130].

The benefit of elevated CSF drug levels becomes apparent when IT and IV treatments are combined, potentially leading to slightly higher antibiotic concentrations in the CSF compared to IVT therapy alone [59]. Although this complexity presents challenges in pharmacokinetic analysis, it proves advantageous in managing MDR-CNS infections. Extensive research supports the efficacy of combining IT administration of antibiotics, such as colistin, vancomycin, tigecycline, and others, in the treatment of CNS infections, including meningitis and ventriculitis, as well as in addressing MDR *Acinetobacter baumannii* ventriculostomy-related infections, when administered concurrently with IV therapy [40,56,59,128,129,130]. Thus, the integration of systemic antibiotic therapy, whether using the same antimicrobial agent or a different active ingredient, is strongly advocated.

Upon examining CSF concentrations following IT administration, the consensus is that the influence of plasma concentrations on CSF kinetics can be overlooked. This simplification is accepted due to the scarcity of studies involving human subjects treated exclusively with IVT anti-infectives [40]. However, the impact of simultaneous IV dosing on CSF-PK for all antibiotics remains uncertain [40,52]. While most patients receive a combination of IV and IT antibiotic doses, the absence of data exclusively focused on IT drug delivery underscores the imperative need for combining IT and systemic (IV) therapy to achieve optimal therapeutic effects [40]. A review paper highlighted that the IT/IVT administration of antibiotics plus IV in the treatment of postsurgical meningitis or ventriculitis decreased mortality 7.09 times compared to IV therapy alone (OR 0.27 [95% CI 0.15–0.49] *p* ≤ 0.00001) with low heterogeneity (Chi^2^  =  7.2 df  =  7, I^2^  =  3%), with no significant differences in reinfection rate or poor functional outcome [127]. The suitable volume ranges for IT injection ranged from 0.5–5 mL [131]. IT and IVT administration of antibiotics have been documented for over a century [132], emphasizing the key advantage of direct antibiotic delivery, bypassing the BBB, leading to higher effective concentrations at the infection site (CNS) and potentially minimizing systemic side effects [40,133,134,135,136]. IT administration injects substances into the thecal sac with CSF, ensuring high therapeutic CNS concentration while minimizing off-target exposure and toxicity [18].

Regarding the uniformity of drug distribution within the CSF following IT administration, we have cited studies that highlight challenges in achieving adequate antibiotic concentrations in the ventricles. The distribution of drugs in the CSF following IT (Intra-lumbar) administration compared to IVT administration is subject to debate. One study noted that drug distribution after intralumbar dosing is not homogeneous and often fails to attain adequate antibiotic concentrations in the ventricles [44].

#### 4.2.3. TDM and Adverse Effect of IT/IVT Drug Administration

The importance of TDM in CSF for patients who exhibit slow response, treatment failure, or risk of drug interactions is emphasized. The decision-making process for performing TDM was influenced by factors such as limited understanding of PK-PD relationships and desired target concentrations in CSF [52]. Despite the potential benefits of TDM, the text acknowledges several challenges, including limited data on CSF drug levels, the need for individualized monitoring, and the risks associated with frequent CSF sampling, such as infections related to ventriculostomy [40,137,138,139]. TDM might be considered in specific cases, like when there is persistent positivity in CSF cultures and evidence of therapeutic failure [40,140]. The proposal to use positron emission tomography for tracking drug distribution in TBM patients indicates a potential shift for future research [92,141].

The risks and adverse effects associated with the procedure of IT/IVT administration of a drug, while minimally invasive, are not negligible. The side effects following IT/IVT dosing of other antibiotics for treatment of Gram-negative meningitis (ventriculitis/meningitis) by gentamycin, polymyxin-B, colistin, tobramycin, AMI, etc. have been observed. The common side effects include chemical ventriculitis, meningitis, seizures, local events, or infection [20,21]. A comprehensive meta-analysis involving 23 studies (229 patients) reported a 13% overall complication rate, with chemical meningitis and seizures as predominant complications [21]. The reported complication rates may not show the true number of issues because patients getting IT treatment are usually in critical health. However, similar adverse effects are also seen with IV dosing alone, without IT administration [142]. Side effects associated with IT therapy may encompass aseptic meningitis, seizures, nerve root irritation, and even brain herniation [125], emphasizing the importance of careful consideration of seizures, chemical ventriculitis, and drug-related toxicities [143]. Although drawbacks associated with these administration routes encompass limited data for determining optimal dosage, the risk of infection due to access to the subarachnoid space and ventricles, and adverse events and risks specific to IT administration [144,145], the IT/IVT delivery of antibiotics has demonstrated enhanced survival and cure rates in individuals with postsurgical meningitis or ventriculitis [127].

### 4.3. Strategies for Prolonged IT DDS in TBM

While the efficacy of IT or IVT administration of RIF and INH has been substantiated by the literature we have reviewed, the frequent injection of drugs via IT routes is invasive and may result in patient non-compliance and procedure-associated risk. In light of this, we have explored prolonged drug delivery methods for IT administration and suggest that utilizing an implantable osmotic and infusion pump or nanoparticle-laden hydrogel could be advantageous for achieving prolonged drug delivery to the CNS. IT-prolonged drug delivery facilitates the direct administration of drugs, enhancing drug concentration in the CNS. Implantable pumps (like Duro’s technology) or NP-laden hydrogel are recommended for sustaining antibiotic levels in the CSF, minimizing the need for frequent injections. They have demonstrated efficacy in delivering medications subcutaneously, intrathecally, and via other routes, for managing prostate cancer, pain, and various medical conditions. Given our understanding that osmotic pumps and NP-laden hydrogels serve as means of prolonged drug delivery and have already been applied for drug delivery in humans, we believe that they could offer potential benefits for the IT delivery of anti-TB drugs.

#### 4.3.1. Implantable Device for Prolonged Drug Delivery

Implantable infusion pumps, utilized for prolonged drug delivery directly into the body, varied in their operation based on the underlying pumping mechanism. These devices were divided into two main categories: passive pumps, which include osmotic pumps and those driven by fluorocarbon propellants, and active pumps, which are typically powered electrically. An ideal implantable drug delivery pump was designed to maintain precise delivery rates over its operational lifespan, fulfilling both standard and increased drug delivery requirements as determined by clinical conditions. It should support various delivery modes, including bolus, continuous, and adjustable-rate delivery, to accommodate different clinical needs. While both implantable osmotic pump devices and battery-powered programmable implantable drug infusion devices were employed for IT drug delivery, they were based on distinct operational principles and possess unique features, making them suitable for various therapeutic applications [146].

Battery-powered programmable implantable drug infusion devices for IT drug delivery: A sophisticated implantable programmable drug delivery device for IT usage represents a significant advancement in targeted drug delivery, particularly for delivering medications directly into the space surrounding the spinal cord. This technology enables precise medication delivery to CNS, significantly reducing the potential for systemic side effects. The device consists of a drug reservoir and a programmable pump, both of which are surgically implanted under the skin. It is connected by a catheter to the IT space, where it can release specific doses of medication at predetermined intervals. This method is especially beneficial for managing chronic conditions such as severe spasticity, chronic pain, and certain types of cancer pain, as it ensures consistent medication levels, minimizes side effects, and enhances the quality of life for patients. Such technology marks a leap forward in personalized medicine and targeted drug delivery, offering new hope for patients with conditions that are challenging to manage with traditional treatment methods. Commercially available drug infusion pumps, including the Medtronic SynchroMed II, Medtronic IsoMed, Codman 3000, and Codman Midstream, provide a range of specifications designed to meet specific medical requirements. For instance, the SynchroMed™ II infusion pump is a battery-powered, programmable, implantable device that delivers drugs to the IT space via an implanted catheter. These pumps can be programmed and managed using the Control Workflow approach, a strategy aimed at reducing the need for systemic opioids and achieving effective pain relief, highlighting the personalized and efficient nature of this DDS [147].

Implantable osmotic pump devices for IT drug delivery: Osmotic pumps are a form of membrane-controlled release DDS, driven by osmotic pressure to administer medication at a steady rate. The core principle of their operation involves water permeating through a semipermeable membrane into the pump. This action allows water to dissolve the drug formulation inside the pump while keeping the active agent itself from passing through the membrane. As the internal solution becomes diluted, osmotic pressure builds up, pushing the dissolved content out of the pump and into the body, thus delivering the active agent at a controlled rate [148]. The delivery rate of osmotic pumps is constant and determined by the device’s design before implantation; however, unlike some other DDS, it cannot be adjusted or programmed after the pump has been implanted. This characteristic makes osmotic pumps suitable for applications where a steady, continuous drug release is required over a specific period. The development of the osmotic pump for drug delivery dates back to 1955, involving a design with three chambers primarily intended for pharmacological research, though it was never patented. The Rose–Nelson pump, consisting of three chambers, was designed for pharmacological research but was never patented [148]. An osmotic pump is recognized as a reliable means of controlled drug delivery, dispensing active agents through osmotic pressure [146]. The pump offered predictable release rates independent of the active agent, suitable for various therapeutic agents. Typically, these pumps exhibit a zero-order release profile following an initial lag, capable of achieving higher release rates compared to conventional diffusion-based systems [149]. The implantable osmotic pump systems come in various designs, such as the Elementary Osmotic Pump (EOP), Rose–Nelson Pump, Push–pull, Push–stick, Controlled Porosity, and Single Composition Osmotic Tablet. Among these, the most commonly utilized is the EOP [146].

Several implantable osmotic pumps for DDS have been developed, with two products from a company gaining popularity among others. The preclinical implantable osmotic pump, developed by Alza and intended for preclinical implantation in laboratory animals, has been used across various anatomical sites and species. Documented in over 10,000 scientific publications, its application has enabled the exploration of innovative therapies with potential implications for human treatment [150]. The ALZET^®^ system has delivered a broad spectrum of active agents, ranging from amino acids to gastrointestinal modulators (e.g., amino acids, anesthetics, antibiotics, antibodies, anticancer agents, anticoagulants, anti-epileptics, antigens, antihypertensives, anti-parasitic agents, anti-Parkinson agents, ascitic fluid, catecholamines, chelators, cholinergic, CNS acting agents, enzyme inhibitors, and gastrointestinal modulators) [148]. Currently, the zero-order DDS osmotic implantable pump, developed by Alza and known as the DUROS^®^ system, represents a significant breakthrough as the first implantable osmotic pump used in humans. The implantable pump, with its compact design incorporating a titanium alloy cylinder and a semi-permeable membrane activated by body fluids, facilitates continuous, zero-order drug release. In vitro studies have highlighted its capability to provide consistent zero-order release for up to one year, demonstrating its potential for long-term therapeutic applications [148]. This pump technology has been widely investigated for its use in various human therapies, offering delivery durations ranging from several months up to a year, making it particularly effective for potent drugs that require precise, extended release. A notable example is the Viadur^®^ system, which employs DUROS technology and has received FDA approval for the continuous delivery of leuprolide in the treatment of advanced prostate cancer, achieving a 12-month release cycle at a daily rate of 125 μg [149,151,152,153].

Development is ongoing for other systems such as the Chronogesic^®^ system for sufentanil delivery in chronic pain management, the Omega system for omega interferon in hepatitis C treatment, and catheterized osmotic pump systems for IT opioid delivery and local chemotherapy for brain tumors [150]. The technology is versatile, capable of administering various compounds irrespective of the drug’s properties, from the leuprolide implant for prostate cancer to salmon calcitonin for osteoporosis and Paget’s disease, typically implanted subcutaneously in the upper arm. This ensures a constant osmotic gradient for effective drug delivery over periods ranging from 3 to 12 months, significantly overcoming the limitations associated with bolus injections [154,155]. The pump’s mechanism relies on osmosis to draw water through a semi-permeable membrane, which expands the osmotic agent, displaces a piston, and dispenses precise drug quantities [155]. User satisfaction is high, with a 96% reimplantation rate and more than 90% of users reporting satisfaction with the comfort and convenience of the device at 24- and 52-week intervals post-implantation [156].

#### 4.3.2. Application of NP-Laden Hydrogel for IT-Prolonged Drug Release

The application of NP-laden hydrogels for IT-prolonged drug release has gained significant attention for its potential to enhance the delivery of nanomedicines in both preclinical and clinical settings. Notably, liposomal cytarabine (DepoCyt^®^) was approved by the FDA in 1999 for the IT treatment of lymphomatous meningitis and has shown effectiveness in various cancer treatments, demonstrating improved PK over free drug administration with cytotoxic levels maintained in the CSF for up to 14 days post-IVT injection and nearly 7 days after IT lumbar injection [19,157,158,159,160]. Furthermore, the transition of several IT medications, including morphine, ziconotide, baclofen, and nusinersen, to clinical trials or FDA approval underscores the significance of nanoparticle-based drug delivery in the subarachnoid space [161]. Additionally, NP-laden hydrogels have gained attention for their potential to improve drug release kinetics and enhance bioavailability [162]. These NP-laden hydrogels, which incorporate nanoparticles within 3D polymer matrices, offer targeted and controlled drug release to specific sites, such as the injured CNS [163,164,165,166]. The broad spread of NPs along the neuraxis underscores their pivotal role in efficient nanomedicine delivery [167]. This technology, incorporating NPs into hydrogels, addressed issues like burst release and low encapsulation efficiency, presenting a promising avenue to improve drug delivery efficiency [162]. Despite their potential and promising preclinical studies, the clinical adoption of NP-laden hydrogels is still emerging, with ongoing research focused on translating experimental findings into clinical practice [168]. Studies have demonstrated the efficacy of thermosensitive gels in delivering drugs like amphotericin-B, reducing dosing frequency and neurotoxicity while enhancing antifungal efficacy in animal models [169].

The availability of IT drug delivery via osmotic pump therapy or NP-hydrogel composite may vary slightly between first-world and third-world countries. While the cost of osmotic pumps compared to battery-powered pumps is lower, accessibility differs based on healthcare infrastructure and regulatory frameworks. In first-world nations, advanced medical facilities and skilled personnel make this therapy more accessible, whereas third-world countries encounter challenges such as limited healthcare resources, medical professional shortages, and regulatory barriers which may affect access to these treatments.

The evolving landscape of IT anti-TB therapy for TBM is paving the way for the development of optimized therapeutic regimens. This progress is driven by an in-depth understanding of various factors including physiological barriers, PKs, drug properties, administration routes, clinical outcomes, and strategies for prolonged drug delivery. Such comprehensive insights are crucial for enhancing the efficacy and safety of treatments for TBM, ultimately improving patient outcomes.

## 5. Conclusions

Understanding the PK-PD parameters of anti-TB drugs is key to optimizing dosages for TBM treatment. Despite facing challenges, IT administration of RIF and INH promising for increasing CSF drug concentration and enhancing clinical outcomes while minimizing systemic side effects. The efficacy and safety of IT/IVT-RIF and INH are supported by case reports, cohort studies, and the literature, even in refractory TBM cases. Osmotic pumps offer a practical solution for sustained IT drug delivery, and nanoparticle-laden hydrogels hold potential for the future, awaiting clinical validation.

Based on CSF-PK data following oral and IV administration of anti-TB drugs, a daily release of 5 mg of RIF and 20 mg of INH via an IT osmotic pump, supplemented by an oral regimen, may effectively eradicate MTB and yield positive clinical outcomes. This approach aims to prevent neurological damage caused by the infection. Despite the hurdles in dose optimization, the current literature supports the concept of optimized regimens for TBM treatment, illustrating a hopeful path forward in the fight against this severe condition.

## Figures and Tables

**Figure 1 pharmaceutics-16-00540-f001:**
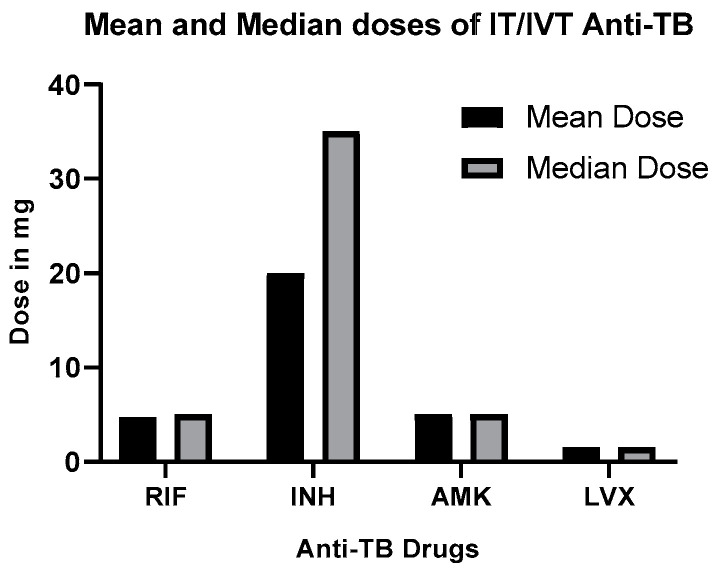
As demonstrated, the mean dose of IT/IVT-RIF is: 4.71 mg/day, INH: 20 mg, LVX: 5 mg, AMK: 1.5 mg, and the median dose of IT/IVT-RIF: is 5 mg, INH: 3 mg, LVX: 5 mg and AMK: 1.5 mg.

**Table 1 pharmaceutics-16-00540-t001:** Systemic therapeutic regimen of anti-TB drugs used in TBM [11,91], WHO Guidelines.

Category	Drug	Adult Dose (WHO) in (mg/kg)	Child Dose (WHO) in (mg/kg)	Duration (WHO)	CSF Penetration (CSF/Plasma%)	Adverse Effects
First-line drugs for DS-TBM	RIF	15 (10–20); max. 600 mg	10 (8–12); max. 600 mg	12 months	10–20%	Hepatotoxicity, orange urine, many drug interactions
	INH	10 (7–15); max. 300 mg	5 (4–6); max. 300 mg	12 months	80–90%	Hepatotoxicity, peripheral neuropathy, lupus-like syndrome, confusion, seizures
	PZA	35 (30–40) mg/k	25 (20–30) mg/kg	First 2 months	90–100%	Hepatotoxicity, arthralgia, gout
	ETB	20 (15–25) mg/kg	15 (15–20) mg/kg	First 2 months	20–30%	Dose-related retrobulbar neuritis, more common in renal impaired
	STM	15–30; max. 1 g IV/ IM	15 (12–18); max. 1 g	First 2 months	10–20%	Nephrotoxicity and ototoxicity
Core second-line drugs for MDR-TBM	LVX	<5 Ys: 15–20, ≥5 Ys: 10–15	10–15 (mg/kg)	During treatment	70–80%	Nausea, headache, tremor, confusion, tendon rupture (rare)
	MXF	10–20: max. 400 mg NWE	400 mg/d	During treatment	70–80%	Nausea, headache, tremor, confusion, tendon rupture (rare)
	AMK	15–30 mg/kg; max. 1 g IV or IM	15; max. 1 g IV or IM	Intensive phase only	10–20%	Nephrotoxicity and ototoxicity
	KAN	15–30 mg/kg; max. 1 g IV or IM	15; max. 1 g IV or IM	Intensive phase only	10–20%	Nephrotoxicity and ototoxicity
	CM	15–30 mg/kg; max. 1 g IV or IM	15; max. 1 g IV or IM	Intensive phase only	No Data	Nephrotoxicity and ototoxicity
	ETO	15–20 mg/kg; max. 1 g	15–20; max. 1 g	During treatment	80–90%	Anorexia, nausea, vomiting, gynaecomastia, hypothyroidism, confusion, seizures
	CYC	10–20 mg/kg; max. 1 g	10–15; max. 1 g	During treatment	80–90%	CNS toxicity
	LNZ	10 mg/kg; max. 600 mg	600 mg/d	During treatment	30–70%	Myelosuppression, optic neuropathy; use with pyridoxine.
Other drugs for MDR-TB, in TBM	CFZ	1 mg/kg	100–200 mg/d	NA	No Data	skin discoloration (orange/red) and sun sensitivity.
		100 mg/d	-	NA	0 [92]	NA
	PAS	200–300 mg/kg	8–12 g	NA	No Data	Vomiting, diarrhea, reversible hypothyroidism (increased risk with ethionamide)
	BDQ	Not determined	400 mg 1x/d for 2 wks, then 200 mg 3x/wks for 22 wks	NA	No Data	Nausea, vomiting, arthralgia, QT prolongation
	Dlm	Not determined	200 mg	NA	No Data	Nausea, vomiting, and dizziness rarely; QT prolongation

Abbreviations: PAS, Para-aminosalicylic acid; RIF, Rifampin; INH, Isoniazid; RFB, Rifabutin; LVX, Levofloxacin; AMK, Amikacin; CYC, Cycloserine; LNZ, Linezolid; CFZ, Clofazimine; BDQ, Bedaquiline; Dlm, Delamanid; CM, Capreomycin; KAN, Kanamycin; ETO, Ethionamide; NEW, not well established; DS, drug sensitive; Ys, years, wks, weeks.

**Table 2 pharmaceutics-16-00540-t002:** Anti-TB PK in serum/CSF, systemic administration for TBM.

Drug	Adm. Route	Sample #	Dose	Pathogen Type and Drug Susc.	MIC in µg/mL	Ser Conc. in µg/mL (Mean ± SD)	CSF Conc. in µg/mL (Mean ± SD)	T_max_ in CSF	CSF/Serum Ratio	References
RIF.	PO	1	NA	INH-R, BDP RIF-R	1	NA	NA	NA	NA	[103]
	PO	19	10.7 ± 0.5 mg/kg	NA	NA	11.5 ± 1.0	0.39 ± 0.06	NA	0.04 ± 0.01	[88]
	PO	10	11.1 ± 0.5 mg/kg	NA	NA	10.6 ± 1.4	0.38 ± 0.06	NA	0.04 ± 0.01	[88]
	PO	7	10.1 ± 0.6 mg/kg	NA	NA	10.1 ± 1.1	0.78 ± 0.13	NA	0.08 ± 0.02	[88]
	PO	7	10.5 ± 0.8 mg/kg	NA	NA	4.7 ± 0.6	0.47 ± 0.06	NA	0.11 ± 0.03	[88]
	PO	26	450 mg/d	NA	NA	C_max_ (6.3 mg/L)	C_max_ (0.21 mg/L)	2	NA	[84]
	IV	26	600 mg/d	NA	NA	C_max_ (22.1 mg/L)	C_max_ (0.60 mg/L)	2	NA	[84]
	PO	1	NA	Pre-XDR-TBM, R	>16	NA	NA	NA	NA	[92]
	In Vitro	-	NA	S. Strain	0.2–0.4	NA	NA	NA	NA	[68]
	PO	20	NA	NA	NA	9.4 (2.9–23.7)	0.2 (0.1–0.4) CCSF_0–2_	NA	NA	[61]
	PO	20	NA	NA	NA	9.4 (2.9–23.7)	0.4 (0.1–1.4) CCSF_0–8_	NA	NA	[61]
INH	PO	1	NA	Mod. RS to INH	2–4	NA	NA	NA	NA	[103]
	PO	19	8.5 ± 0.4 mg/kg	NA	NA	4.4 ± 0.5	1.9 ± 0.3	NA	0.47 ± 0.04	[88]
	PO	8	9.1 ± 0.6 mg/kg	NA	NA	2.6 ± 0.8	3.2 ± 0.8	NA	1.31 ± 0.13	[88]
	PO	9	9.0 ± 0.8 mg/kg	NA	NA	2.1 ± 0.6	1.8 ± 0.5	NA	1.03 ± 0.14	[88]
	PO	8	7.5 ± 0.9 mg/kg	NA	NA	1.0 ± 0.3	1.8 ± 0.5	NA	2.12 ± 0.25	[88]
	PO	1	600 mg/d	Pre-XDR-TBM, R	0.06	9.86 C_max_	11.57 C_max_	NA	1.51	[92]
	In Vitro		NA	S. Strain	0.02–0.04	NA	NA	NA	NA	[68]
	PO	6 Child	2.5–3.3 mg/kg tid–qid	S. Strain	-	4.84 ± 2.31	3.18 ± 1.27	-	-	[65,104]
	PO	20	NA	NA	NA	4.6 (1.0–10.0) 4.7 (2.5–13.6)	1.4 (0.5–6.1) 1.3 (1.2–4.3)	NA	NA	[61]
RFB	PO	1	NA	RFB sus.	≤0.250–0.5	NA	NA	NA	NA	[103]
	PO	1	NA	Pre-XDR-TBM, R	8	NA	NA	NA	NA	[92]
LVX	PO	1	10–15 mg/kg	NA	NA	NA	NA	NA	NA	[11]
OFX	In Vitro	-	-	-	(0.5–1.0)	-	-	-	-	[68]
AMK	IV or IM	1	15 mg/kg	NA	NA	NA	10–20% of Ser conc.	NA	NA	[11]
	IV or IM	1	-	-	0.5	-	-	-	-	[92]
Mfx	PO	1	400 mg	NA	NA	NA	70–80% of Ser conc.	NA	NA	[11]
	PO	1	400 mg/d	Pre-XDR-TBM, R	4	2.11	0.62	-	0.48	[92]
CYC	PO	1	10–15 mg/kg	NA	NA	NA	80–90% of Ser Conc.	NA	NA	[11]
	PO	1	500 mg/d	Pre-XDR-TBM, R	16	36.28 C_max_	20.62 C_max_	-	-	[92]
LNZ	PO	1	600 mg	NA	NA	NA	30–70% of Ser conc.	NA	NA	[11]
	PO	14	600 mg 2x/d	NA	NA	18.6 ± 9.6	10.8 ± 5.7	101.6 ± 59.6 μg·h/mL	0.66	[100]
	PO	1	600 mg 2x/d	Pre-XDR-TBM, R	-	31.81	15.72	-	-	[92]
CFZ	PO	1	100–200 mg	NA	NA	NA	Probably low	NA	NA	[11]
	PO	1	100 mg/d	Pre-XDR-TBM, R	-	0.35	0	-	-	[92]
BDQ	PO	7	400 mg 1x/d for 2 wks, then 200 mg 3x/wk for 22 wks.	NA	>30 ng/mL	1.1442 (C_max_ = 1.832)	0.00149 (C_max_ = 0.00379)	5 h	0.12	[101]
	PO	1	400 mg 1x/d for 2 wks then 200 mg 3x/wk for 22 wks.	NA	NA	NA	Probably very low	NA	NA	[11]

Abbreviations: RIF, Rifampin; INH, Isoniazid; RFB, Rifabutin; LVX, Levofloxacin; AMK, Amikacin; CYC, Cycloserine; LNZ, Linezolid; CFZ, Clofazimine; BDQ, Bedaquiline; R, Resistant; Mod., Moderate; Sus., Susceptible; BL, Borderline; d, daily; wks, weeks.

## Data Availability

Not applicable (due to this review summarizing the published literature).

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
