# Peer review of "Comprehensive Therapeutic Approaches to Tuberculous Meningitis: Pharmacokinetics, Combined Dosing, and Advanced Intrathecal Therapies"

_pharmaceutics, 2024, doi:10.3390/pharmaceutics16040540_

Round 1

Reviewer 1 Report

Comments and Suggestions for Authors

This review article entitled “Comprehensive Therapeutic Approaches to Tuberculous Meningitis: Pharmacokinetics, Combined Dosing, and Advanced Intrathecal Therapies” by Madadi and Sohn gives an important overview of current therapeutic approaches of TBM. The review is constructed in a very well-organized manner: The introduction begins with the seriousness of the TBM, which yields high mortality rate especially among the compromised juveniles. It then explains why it is difficult to treat specifically TBM with conventional TB treatments, leading the readers to grasp the importance of IT or IVT therapy. The content of this review is well supported by the literature, and the writing is logic and clear. It will certainly be a great interest to the readership of the journal Pharmaceutics. There are only a few suggestions for improvement as described below:

Although it is inevitable for this type of articles, there are numerous abbreviations used in this article. There is a list attached to Table2, but it is helpful to have a list for the entire text.

In addition, the definition should be present when the abbreviation first appears. Please define the following:

L337    MIC and AUC

In L83, the authors mentioned the problem of multi-drug-resistant strain. It is a well-known phenomenon that affects the treatments of many diseases. Are there any strategies taken for the MDR-TBM? Or any suggestions? This paragraph seems to end with an especially negative note.   

Author Response

Thank you for your valuable feedback regarding the use of abbreviations and the need for their definition in our manuscript. We understand the importance of clarity and consistency in scientific writing and have taken steps to address your concerns.

Firstly, we have included a comprehensive list of abbreviations (pages 38 and 39) used throughout the entire text to aid readers' understanding. Additionally, we have ensured that each abbreviation is defined upon its first appearance in the manuscript, including MIC and AUC as requested (L337).

Regarding the paragraph discussing multi-drug-resistant strain implications (L83), we appreciate your observation and acknowledge the need to provide potential strategies or suggestions to mitigate the negative impact. To address this, we have added a sentence (L86, 87, and 88) suggesting the critical need for early detection of drug resistance, customized treatment regimens, and enhanced anti-TB drug concentration in CSF, to effectively manage MDR-TBM.

We hope these revisions enhance the clarity and readability of our manuscript, and we remain open to further suggestions or feedback.

Reviewer 2 Report

Comments and Suggestions for Authors

The work by Madadi and Sohn summarises the therapy of TBM very comprehensively. On the one hand, the pharmacokinetics and, on the other, cases are discussed and evaluated in detail. Overall, a very good summary of the current state of therapy. It would be advantageous for the flow of reading and understanding if the sections were kept smaller. Perhaps one or two illustrations would also help to make the manuscript easier to understand. For example, how do the bacteria get into the CNS? The point of the blood-brain barrier in particular does not really seem to be clear, although it is certainly one of the most important points for the success of the therapy. This should perhaps be looked at more closely. Is the possibility of therapy also a question of the country? In other words, first and third world?

Author Response

Thank you for your thoughtful feedback on our manuscript. We greatly appreciate your positive assessment of the comprehensive coverage of pharmacokinetics and detailed case evaluations.

We have added paragraphs (under the section of the introduction, subsection 1.1 entitled Physiology of BBB and CSF, page 3) elaborating on the role of the blood-brain barrier (BBB) and its significance in therapy success, recognizing its pivotal role in treatment outcomes.

Regarding the possibility of therapy varying by country, we have included a paragraph exploring this aspect (Page 26). We discuss whether therapy options differ between first and third-world countries, considering factors such as access to resources and healthcare infrastructure. This addition aims to provide a broader perspective on therapeutic approaches to TBM.

Regarding the addition of illustrations in the manuscript, I would like to emphasize that our primary focus in this manuscript is on the treatment of TBM and the eradication of the Blood-Brain Barrier (BBB). However, including an illustration to demonstrate how bacteria can spread to the Central Nervous System (CNS) may deviate slightly from our main focus.

We believe these revisions address your concerns and contribute to enhancing the clarity and comprehensibility of our manuscript. We remain open to further suggestions and appreciate your valuable input in improving our work.

All sentences, paragraphs, and changes made to the manuscript are highlighted in yellow.
